# Inhibition of the serine protease HtrA1 by SerpinE2 suggests an extracellular proteolytic pathway in the control of neural crest migration

Edgar M Pera[1]*, Josefine Nilsson-De Moura[1], Yuriy Pomeshchik[2†], Laurent Roybon[2†], Ivana Milas[1]

[1]Vertebrate Developmental Biology Laboratory, Department of Laboratory Medicine, Lund Stem Cell Center, University of Lund, Lund, Sweden; [2]iPSC Laboratory for CNS Disease Modeling, Department of Experimental Medical Science, Lund Stem Cell Center, Strategic Research Area MultiPark, Lund University, Lund, Sweden

**Abstract** We previously showed that SerpinE2 and the serine protease HtrA1 modulate fibroblast growth factor (FGF) signaling in germ layer specification and head-to-tail development of *Xenopus* embryos. Here, we present an extracellular proteolytic mechanism involving this serpin-protease system in the developing neural crest (NC). Knockdown of SerpinE2 by injected antisense morpholino oligonucleotides did not affect the specification of NC progenitors but instead inhibited the migration of NC cells, causing defects in dorsal fin, melanocyte, and craniofacial cartilage formation. Similarly, overexpression of the HtrA1 protease impaired NC cell migration and the formation of NC-derived structures. The phenotype of SerpinE2 knockdown was overcome by concomitant downregulation of HtrA1, indicating that SerpinE2 stimulates NC migration by inhibiting endogenous HtrA1 activity. SerpinE2 binds to HtrA1, and the HtrA1 protease triggers degradation of the cell surface proteoglycan Syndecan-4 (Sdc4). Microinjection of *Sdc4* mRNA partially rescued NC migration defects induced by both HtrA1 upregulation and SerpinE2 downregulation. These epistatic experiments suggest a proteolytic pathway by a double inhibition mechanism:

SerpinE2 ⊣ HtrA1 protease ⊣ Syndecan-4 → NC cell migration.

*For correspondence:
edgar.pera@med.lu.se

Present address: †Department of Neurodegenerative Science, The MiND Program, Van Andel Institute, Grand Rapids, Michigan, United States

Competing interest: The authors declare that no competing interests exist.

## eLife assessment

This **fundamental** work substantially advances our understanding of cell migration, especially in that of cranial neural crest. The additional evidence provided to support the conclusion is **exceptional**, with rigorous biochemical assays for materials used and with intensive genetic interventions. The work will be of broad interest to developmental biologists and cell biologists.

## Introduction

Collective cell migration is a fundamental process in the development and maintenance of multicellular organisms (*Friedl and Gilmour, 2009*; *Shellard and Mayor, 2020*). Embryonic development relies on the coordinated movement of cells to specific locations, and aberrant cell migration is linked to cancer metastasis. The acquisition of cell motility is associated with an epithelial-mesenchymal transition (EMT), in which cells disrupt epithelial adhesions and remodel junctional complexes in favor of cell-matrix adhesions to adopt a migratory behavior (*Nieto et al., 2016*; *Piacentino et al., 2020*). The EMT does not have to be complete, and cells can be motile while maintaining contact with one

another. A prime example for the study of collective cell migration is the neural crest (NC) in verte-brate embryos (*Szabó and Mayor, 2018*). The NC is a multipotent cell population that is specified at the border of the neural plate (*Le Douarin and Kalcheim, 1999*; *Schock et al., 2023*). EMT initiates migration streams of NC cells toward their targets where they differentiate into diverse cell types and tissues, including peripheral nervous system, melanocytes, and craniofacial skeleton. The behavior of NC cells recapitulates certain stages of cancer progression and metastasis (*Nieto et al., 2016*). Defects in NC development can lead to many congenital syndromes and tumors of the NC lineage (*Medina-Cuadra and Monsoro-Burq, 2021*). These neurocristopathies highlight the need to better understand the molecular basis of key processes in NC development, including EMT and collective migration.

HtrA1 belongs to a conserved family of serine proteases that are homologous to the heat shock-induced HtrA (<u>h</u>igh <u>t</u>emperature <u>r</u>equirement <u>A</u>) peptidase from bacteria and primarily involved in protein quality control and degradation (*Zurawa-Janicka et al., 2017*). Vertebrate HtrA proteases, comprising the four members HtrA1–4, share a trypsin serine protease domain and a carboxyterminal PDZ domain with their bacterial counterpart. The HtrA family is implicated in various pathological conditions including cancer, arthritis, neurodegenerative diseases, and pregnancy disorders. HtrA1 modulates the extracellular matrix and cell signaling as a secreted protein but was found to be active in the cytoplasm and nucleus, too (*Clawson et al., 2008*; *Chien et al., 2009a*; *Campioni et al., 2010*). The function of HtrA1 in cell migration has been studied in vitro to reveal mainly a negative role, although also a positive role has been reported (*Pei et al., 2015*). How HtrA1 activities are regulated is poorly understood, and the mechanism by which this protease affects cell migration in vivo remains elusive.

Members of the serpin superfamily contain a carboxyterminal reactive center loop (RCL) that cova-lently binds to and inhibits target serine proteases inside and outside of cells (*Olson and Gettins, 2011*). Serpin peptidase inhibitor clade E member 2 (SerpinE2), also known as protease nexin-1 (PN1) or glia-derived nexin, has important roles in the nervous, blood, and reproductive systems (*Arocas and Bouton, 2015*; *Monard, 2017*). SerpinE2 also functions as a key factor in tumor dissemination, but the molecular mechanism by which this protease inhibitor governs cell migration and metastasis is largely unknown.

We previously showed that HtrA1 and SerpinE2 are transcriptionally induced by fibroblast growth factor (FGF) signals and act as feedback regulators of FGF/Erk signaling in germ layer and anteropos-terior axis formation in the early *Xenopus* embryo (*Hou et al., 2007*; *Acosta et al., 2015*). HtrA1 acti-vates Erk (extracellular signal-regulated kinase) and expression of the transcription factor Brachyury in the posterior mesoderm, whereas SerpinE2 suppresses Erk activation and *Brachyury* expression in the anterior ectoderm. The HtrA1 protease releases FGF ligands by triggering the cleavage of cell surface proteoglycans such as Syndecan-4 (Sdc4), thereby stimulating FGF/Erk signaling in mesoderm and trunk/tail formation (*Hou et al., 2007*). SerpinE2 binds to and inhibits HtrA1, thus restricting FGF/Erk signaling and allowing ectoderm and head formation to occur (*Acosta et al., 2015*). Since SerpinE2 and HtrA1 exhibit overlapping gene expression in the NC, we now asked whether these proteins might have a role in NC development.

Here, we introduce SerpinE2 as a key player in collective NC cell migration. We show that SerpinE2 promotes NC cell migration via inhibition of the secreted serine protease HtrA1. SerpinE2 de-re-presses the HtrA1-mediated block of NC migration in mRNA-injected embryos, and the ability of this protease inhibitor to rescue cell migration depends on its extracellular location and intact RCL. In epistatic experiments, Sdc4 mRNA can partly revert the NC migration defects that are induced by SerpinE2 knockdown or HtrA1 overexpression. We conclude that the SerpinE2/HtrA1/Sdc4 pathway regulates NC cell migration in the developing embryo.

## Results

### *SerpinE2* and *HtrA1* are expressed in NC cells

The expression of *SerpinE2* and *HtrA1* in early *Xenopus* embryos was previously reported by us and others (*Pera et al., 2005*; *Onuma et al., 2006*; *Hou et al., 2007*; *Acosta et al., 2015*). To investi-gate whether *SerpinE2* and *HtrA1* are expressed in the developing NC, we performed whole-mount in situ hybridization analysis of these genes side by side with the NC marker *Twist* (*Figure 1*). At

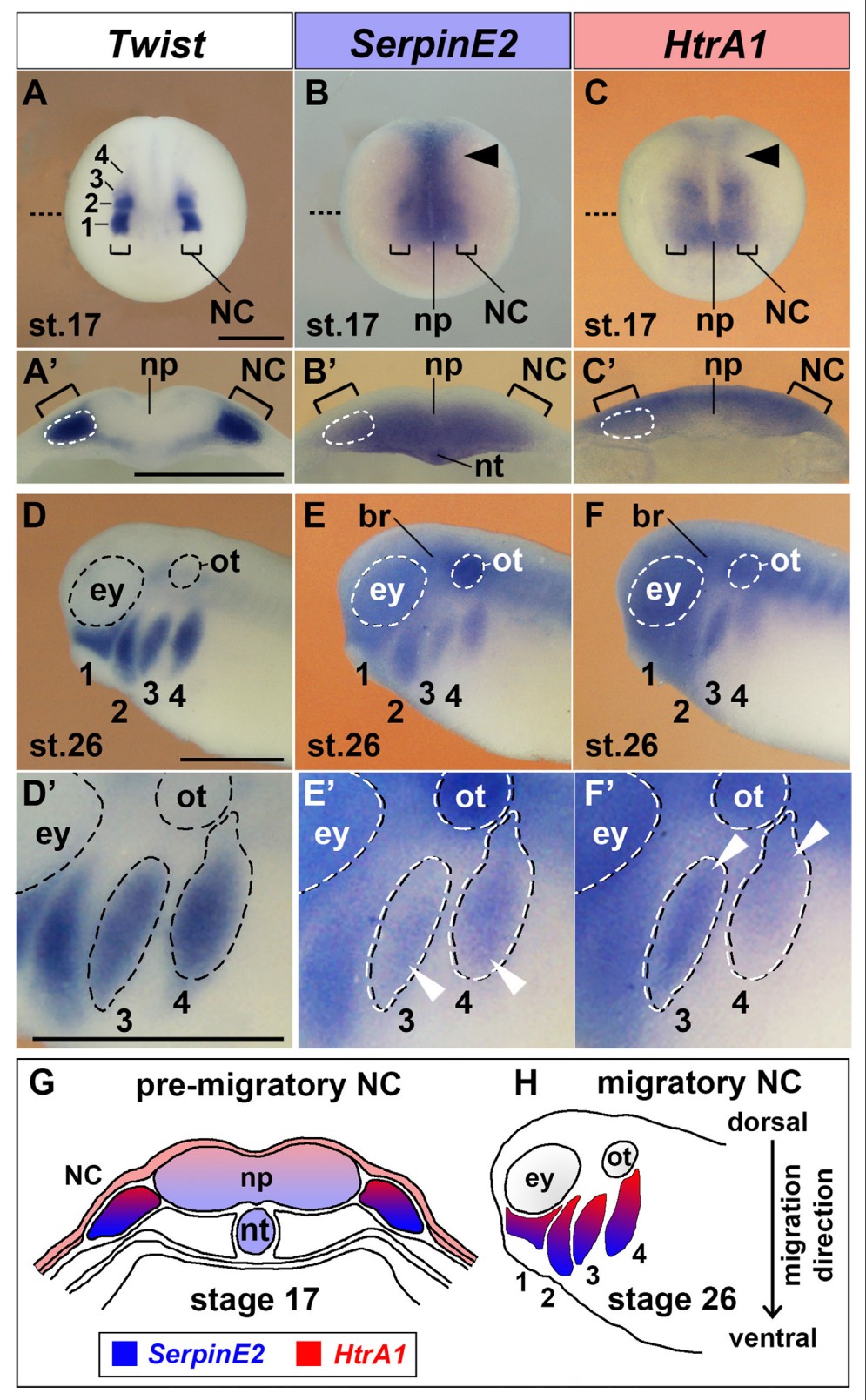

**Figure 1.** SerpinE2 and HtrA1 are expressed in the neural crest. *Xenopus* embryos were analyzed by whole-mount in situ hybridization. (**A–C**) Anterior view of embryos at stage 17. The brackets point to pre-migratory NC cells on each side of the neural plate. The numbers label the *Twist*-expressing cranial NC segments: 1, mandibular; 2, hyoid; 3, anterior branchial; 4, posterior branchial. Arrowheads show *SerpinE2* and *HtrA1* transcripts in the

*Figure 1 continued on next page*

*Figure 1 continued*

trunk NC. The stippled lines indicate the level of sections in A'–C'. (**A'–C'**) Transversally hemisectioned embryos. *SerpinE2* and *HtrA1* signals appear in the NC (stripled circle lines). Note that *SerpinE2* is also expressed in the inner sensorial layer of the neural plate and underlying notochord, whereas *HtrA1* expression is more abundant in the outer ependymal layer of the neural plate. (**D–F**) Lateral view of embryos at stage 26. *SerpinE2* and *HtrA1* are expressed in *Twist*⁺ NC cell streams (1–4). Transcripts of both genes can also be seen in the brain, eye, and otic placode. (**D'–F'**) Magnification of embryos. Arrowheads demarcate *SerpinE2* transcripts near the front (**E'**) and *HtrA1* transcripts at the rear end (**F'**) of the migrating NC cell collectives in the branchial arches. (**G, H**) Summary of gene expression domains. At stage 17, *SerpinE2* is transcribed in ventral and *HtrA1* in dorsal cells of the pre-migratory NC (**G**). At stage 26, *SerpinE2* is expressed in leader cells and *HtrA1* in follower cells of migrating NC streams (**H**).br, brain; ey, eye; NC, neural crest; np, neural plate; nt, notochord; ot, otic placode. Scale bars, 0.5 mm.

The online version of this article includes the following figure supplement(s) for figure 1:

**Figure supplement 1.** *Twist*, *SerpinE2,* and *HtrA1* gene expression in sections of tailbud stage embryos after whole-mount in situ hybridization.

neurula stage 17, *Twist* expression labeled pre-migratory NC cells in the deep layer of the ectoderm (*Figure 1A and A'*; *Hopwood et al., 1989*). *SerpinE2* transcripts were found in the deep layer of the neural plate and adjacent NC cells (*Figure 1B and B'*). *HtrA1* was transcribed in the superficial layer of the neural plate and in the NC (*Figure 1C and C'*). At stage 26, *SerpinE2* and *HtrA1* were co-expressed with *Twist* in ventrally migrating NC cells in the mandibular, hyoid, anterior branchial, and posterior branchial streams (numbered as 1–4 in *Figure 1D–F*; see also *Figure 1— figure supplement 1A–D*). Importantly, *SerpinE2* transcripts accumulated at the ventral leading front, while *HtrA1* expression was abundant in more dorsal follower cells within the migrating NC cohorts (*Figure 1D'–F'*; see also *Figure 1—figure supplement 1E–G*). In addition, *SerpinE2* and *HtrA1* shared overlapping expression in the brain, eye vesicles, and otic placodes (*Figure 1E and F*; see also *Figure 1—figure supplement 1F and G*). These results showed that *SerpinE2* and *HtrA1* are expressed in pre-migratory NC cells and adjacent tissues (*Figure 1G*), and that the SerpinE2 inhibitor transcripts prevail in the leading edge and the HtrA1 protease expression is predominant in the following migratory NC cells (*Figure 1H*).

## SerpinE2 knockdown reproduces the phenotype of NC extirpation in *Xenopus* embryos

Classical extirpation experiments carried out in the urodele *Amblystoma* demonstrated that the cranial NC is important for the formation of the head tissue (*Stone, 1922*; *Stone, 1926*) and the trunk NC for dorsal fin and melanocyte development (*DuShane, 1935*). We reproduced these findings in *Xenopus* and showed that bilateral removal of the neural folds (anlagen of the NC) in the cranial and anterior trunk region of a mid-neurula embryo at stage 17 resulted in larvae displaying a reduced head size, missing dorsal fin (open arrowheads), and reduced melanocyte pigmentation (arrow) (*Figure 2A–C*).

The expression pattern prompted us to investigate the function of SerpinE2 in NC development. *Xenopus laevis* is allotetraploid (*Session et al., 2016*) and contains two *SerpinE2* genes, namely *SerpinE2.L* (*PN1.a*) and *SerpinE2.S* (*PN1.b*), which encode two proteins that share 96% amino acid identity. A combination of two antisense morpholino oligonucleotides (MOs) that target the translation initiation site of *SerpinE2.L* and one MO directed against *SerpinE2.S* (collectively termed *SerpinE2*-MO) efficiently blocks SerpinE2 protein biosynthesis in *Xenopus* embryos (*Acosta et al., 2015*). We previously noted reduction of head structures in tadpoles upon microinjection of *SerpinE2*-MO into the animal pole blastomeres at the eight-cell stage. A closer analysis now revealed that these *SerpinE2*-morphant embryos also exhibited a loss of dorsal fin structures (open arrowheads) and a decreased number of melanocytes (arrow) at stage 40 (*Figure 2E*). These phenotypes were specific, since a standard control-MO had no phenotypic effect (*Figure 2D*), and co-injection of *SerpinE2*-MO together with a *Flag-SerpinE2* mRNA that is not targeted by the MO rescued normal development (*Figure 2F*). The striking similarity of the phenotype obtained after knockdown of SerpinE2 with that of NC extirpation (*Figure 2B and E*) and supporting evidence from fate mapping studies that NC cells contribute to head mesenchyme, dorsal fin structures, and melanocytes (*Tucker, 1986*; *Sadaghiani and Thiébaud, 1987*; *Tucker and Slack, 2004*) suggest that SerpinE2 might be important for NC development.

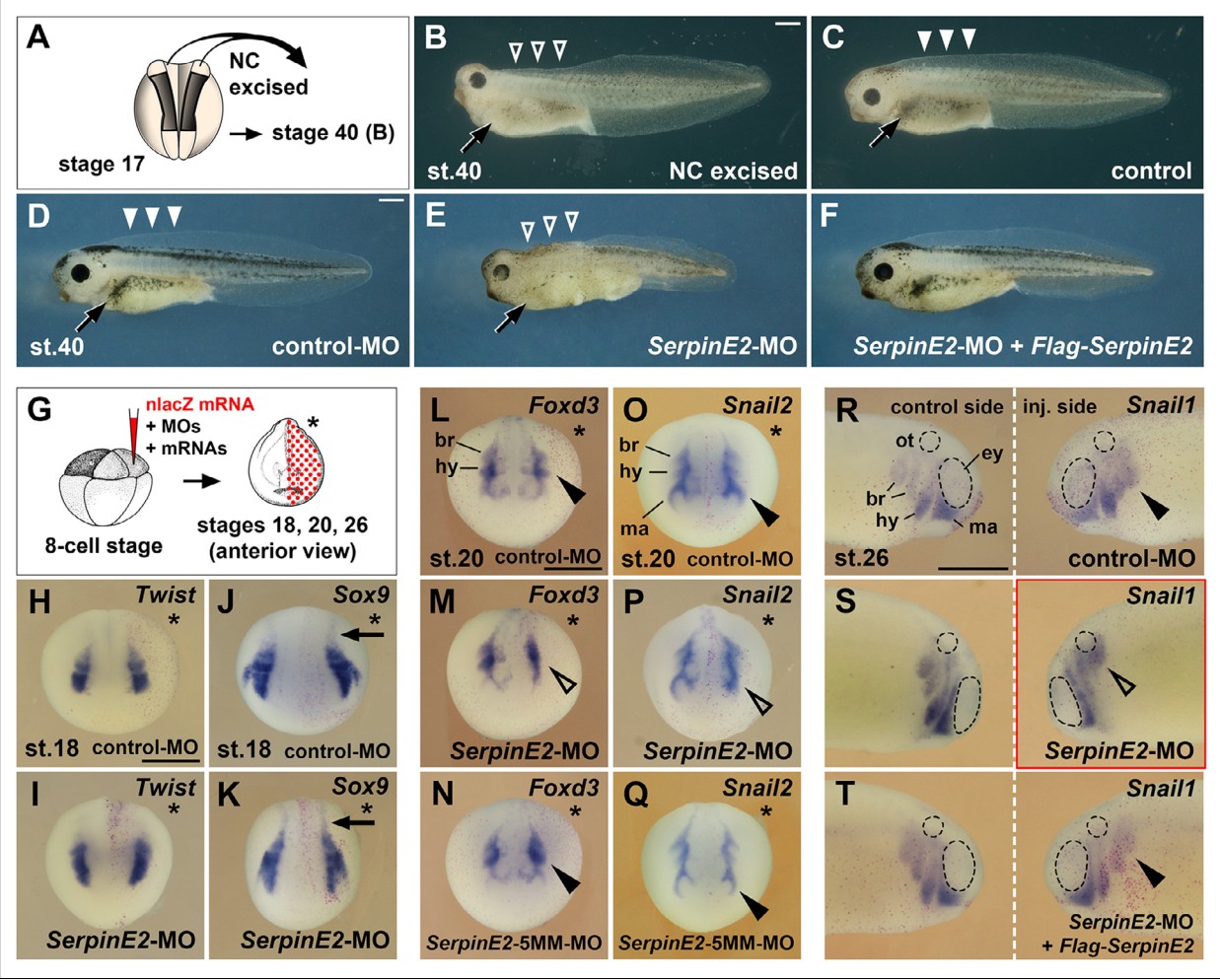

**Figure 2.** Knockdown of SerpinE2 mimics the phenotype of neural excision and inhibits migration of neural crest cells. (**A**) Scheme of extirpation. Dorsal view of *Xenopus* embryo at stage 17, from which NC tissue was removed on both sides. (**B, C**) Tadpole embryo at stage 40 following NC excision (**B**) and sibling control (**C**). Note the small head, absence of dorsal fin tissue (open arrowheads), and the reduced number of melanocytes (arrow) resulting from NC extirpation in A. (**D**) Unaffected tadpole after microinjection with control-MO into all animal blastomeres at the eight-cell stage. Highlighted are the pigmented melanocytes (arrow) and the intact dorsal fin (filled arrowheads). (**E**) *SerpinE2*-MO causes a reduction of head tissue, dorsal fin structures (open arrowheads), and melanocytes (arrow). (**F**) Co-injection of *SerpinE2*-MO and 2 ng non-targeted *Flag-SerpinE2* mRNA restores a normal phenotype. (**G**) Scheme for microinjections in H–T. MOs and mRNAs were injected together with 100 pg *nlacZ* mRNA as lineage tracer (red nuclei) into one dorsal animal blastomere of embryos at the eight-cell stage. The injected side is marked with a star. (**H–K**) Anterior view of neurula embryos at stage 18. Neither control-MO nor *SerpinE2*-MO affect *Twist* and *Sox9* expression in NC cells in the head and trunk (arrows) . (**L–Q**) *SerpinE2*-MO inhibits the epithelial-mesenchymal transition (EMT) of *Foxd3*⁺ and *Snail2*⁺ NC cells (open arrowheads) at stage 20, whereas the control-MO and *SerpinE2*-5MM-MO have no effect (filled arrowheads). (**R–T**) Lateral view of stage 26 embryos. A single injection of control-MO does not affect the migration of *Snail1*⁺ NC cells (filled arrowhead). *SerpinE2*-MO leads to a migration defect on the injected side (open arrowhead). 333 pg *Flag-SerpinE2* mRNA rescues NC migration in the *SerpinE2*-morphant embryo. br, branchial crest segment; ey, eye primordium; hy, hyoid crest segment; ma, mandibular crest segment; MO, morpholino oligonucleotide; NC, neural crest; ot, otic vesicle. Doses of injected MOs per embryo were 40 ng (**D–F**) and 10 ng (**H–T**). Indicated phenotypes were shown in B, 10/11; D, 89/90; E, 64/83; F, 122/132; H, 29/29; I, 24/26; J, 7/7; K, 10/11; L, 9/9; M, 7/7; N, 7/9; O, 7/8; P, 8/9; Q, 9/11; R, 10/10; S, 13/15; T, 9/9. Scale bars, 0.5 mm.

The online version of this article includes the following figure supplement(s) for figure 2:

**Figure supplement 1.** SerpinE2 depletion does not affect the specification of neural crest cells.

## SerpinE2 is dispensable for the specification but essential for the migration of NC cells

We injected MOs into the animal pole of embryos at the eight-cell stage together with *nlacZ* mRNA as lineage tracer (red nuclei) to identify the injected side and ensure that the MO is properly targeted (*Figure 2G*). *SerpinE2*-MO does not appear to affect NC specification in the head and trunk, as

the expression of *Twist*, *Sox9*, *cMyc*, *Foxd3*, *Snail1,* and *Snail2* in pre-migratory NC cells remained unchanged in mid-neurula embryos at stage 18 (*Figure 2H–K*; see also *Figure 2—figure supplement 1*).

Since migration of the NC is initiated progressively from anterior to posterior in the neural folds of the closing neural tube (*Sadaghiani and Thiébaud, 1987*), we chose two distinct stages to monitor the EMT and migration of this cell population. At stage 20, NC cells of the mandibular crest segment are migrating from the mesencephalon around the eye primordium, while cells of the hyoid and branchial crest segments are undergoing EMT in the rhombencephalon (*Figure 2L and O*). At stage 26, NC cell migration occurs in the hyoid segment anterior to the otic vesicle, and in two split branchial segments posterior to the ear primordium (*Figure 2R*). A single dorsal injection of *SerpinE2*-MO caused a delay or failure of these NC cells to undergo EMT and migration (open arrowheads) in advanced neurula (*Figure 2M and P*) and tailbud embryos (*Figure 2S*). These knockdown effects were specific, because a *SerpinE2*-5MM-MO, which contains five mismatches with the *SerpinE2.L* and *SerpinE2.S* target mRNA sequences, as well as a combination of *SerpinE2*-MO and non-targeted *Flag-SerpinE2* mRNA failed to disrupt NC migration (*Figure 2N, Q, and T*; see also Figure 8B–E' and *Figure 8—figure supplement 1A and B*). Ventrally injected *SerpinE2*-MO was less efficient in reducing NC migration (*Figure 8—figure supplement 1C and D*), because expression of the *SerpinE2* is highest in dorsal regions of post-gastrula embryos (*Figure 1B*). We conclude that SerpinE2 is dispensable for the initial specification but necessary for the migration of NC cells in *Xenopus* embryos.

## HtrA1 inhibits the development and migration of NC cells

Next, we investigated whether HtrA1 affects NC development (*Figure 3*). Injection of *HtrA1* mRNA into the animal pole blastomeres led to a decrease of head and eye structures in early tadpole embryos (*Figure 3A and B*) in accordance with the previously reported axis posteriorizing activity of this protease (*Hou et al., 2007*). HtrA1 overexpression also caused a reduction of dorsal fin tissue (open arrowheads) and fewer melanocytes (arrow), suggesting that HtrA1 inhibits the formation of NC-derived structures.

A single dorsal injection of *HtrA1* mRNA did not affect the expression of cranial and trunk NC markers at stage 18 (*Figure 3E and F*) but blocked EMT and migration of NC cells at stages 20 and 26 (*Figure 3H–K* and Figure 6B–C'). Quantitative analysis showed that HtrA1 overexpression caused migratory defects in a concentration-dependent manner (*Figure 6—figure supplement 1A and B*). Ventral mRNA injection of *HtrA1*-derived constructs does not properly target the NC and therefore was less effective in inhibiting the migration of this cell population (*Figure 6—figure supplement 1C and D*).

Using western blot analysis with a polyclonal anti-HtrA1 antibody, we previously showed that a morpholino oligonucleotide directed against *HtrA1.L* and *HtrA1.S*, designated *HtrA1*-MO, efficiently blocks endogenous HtrA1 protein expression in *Xenopus* embryos (*Hou et al., 2007*). Dorsal injection of *HtrA1*-MO failed both to affect the induction of the NC marker *Twist* (*Figure 3G*) and to alter the migration of *Twist*[+] cells (*Figure 3L*), suggesting that knockdown of HtrA1 appears not to affect NC development. We conclude that upregulation of HtrA1 protease activity inhibits the migration but not the specification of NC cells.

## HtrA1 and SerpinE2 control the development of the head cartilage skeleton

Cranial NC cells contribute to the craniofacial skeleton (*Sadaghiani and Thiébaud, 1987*). The mandibular crest stream gives rise to the ethmoid-traberculum (upper jaw), palatoquadrate and Meckel's cartilage (lower jaw); the hyoid crest stream supplies the ceratohyal cartilage; and the branchial crest streams form the paired ceratobranchial cartilages (gills) (*Figure 4L*). The transcription factor Sox9 is a master regulator of chondrogenesis that mediates the condensation of chondrogenic mesenchyme and chondrocyte differentiation (*Suzuki et al., 2012*). In early tadpoles at stages 40/41, *Sox9* expression demarcates chondrogenic precursors in the head mesenchyme (*Figure 4A and A'*). Upon microinjection of *HtrA1* mRNA into the animal pole of all blastomeres at the four-cell stage, *Sox9* expression was reduced in developing cartilage structures (*Figure 4B and B'*). A similar reduction of *Sox9*[+] cartilaginous elements was induced by *SerpinE2*-MO, whereas the control-MO had no effect

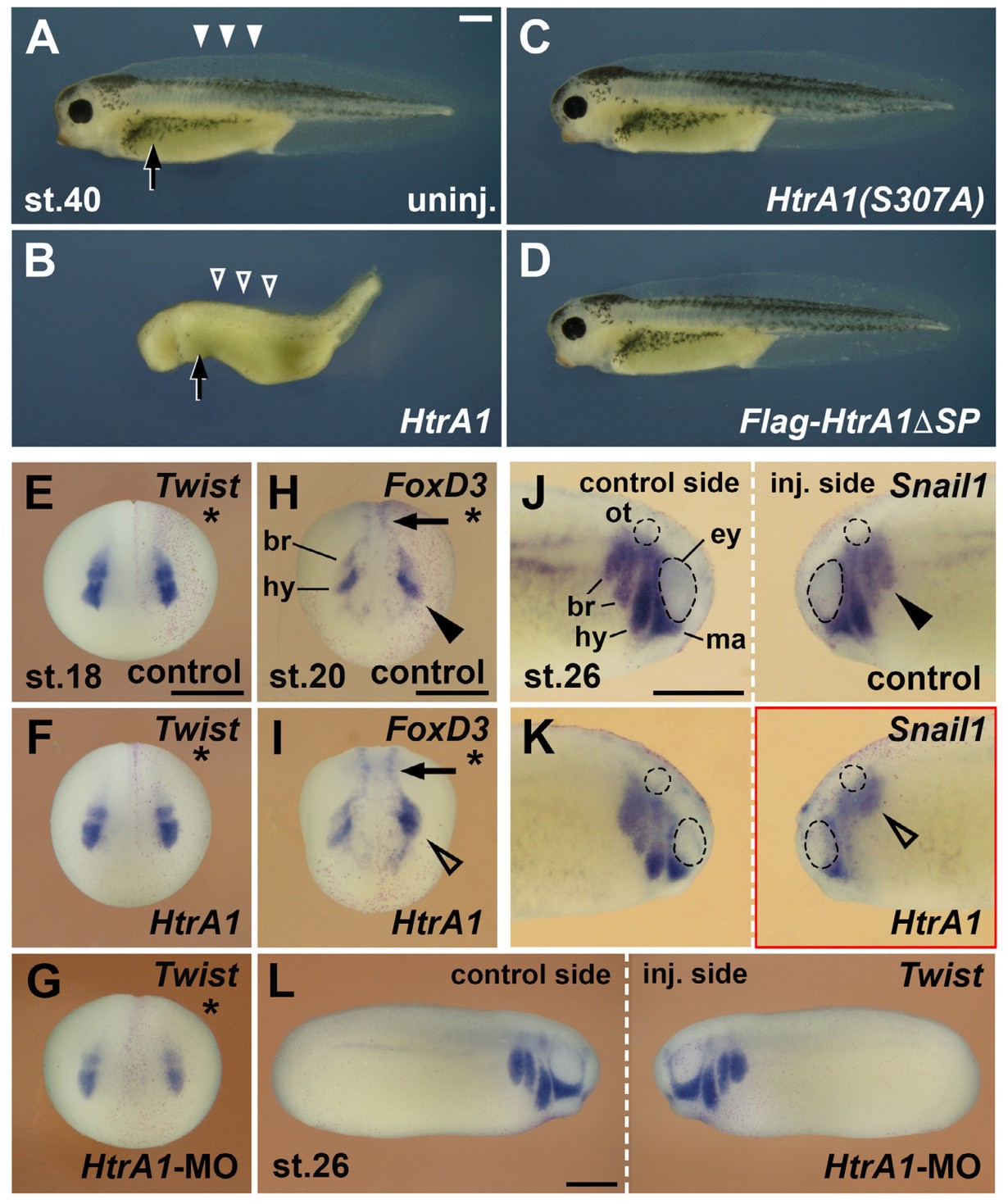

**Figure 3.** HtrA1 protease inhibits the formation of neural crest (NC)-derived structures and reduces NC migration. Embryos were injected into all animal blastomeres (**A–D**) or a single dorsal animal blastomere (**E–L**) at the eight-cell stage. (**A–D**) Tadpoles at stage 40. *HtrA1* mRNA causes reduction of head tissue, dorsal fin structures (arrowheads), and melanocytes (arrow), whereas *HtrA1(S307A)* and *Flag-HtrA1ΔSP* mRNA have no effect. (**E–G**) Anterior view of embryos at stage 18. Stars demarcate the injected sides. Neither 65 pg *HtrA1* mRNA nor 10 ng *HtrA1*-MO do affect the specification of *Twist*+ cranial NC cells. (**H, I**) Anterior view of embryos at stage 20. 65 pg *HtrA1* mRNA reduces the epithelial-mesenchymal transition (EMT) of *Foxd3*+ cranial NC cells (arrowheads) but does not affect the specification of trunk NC cells (arrows). (**J–L**) Lateral view of embryos at stage 26. The migration of NC cells (arrowheads) is reduced by 65 pg *HtrA1* mRNA but not affected by 10 ng *HtrA1*-MO. br, branchial crest segments; ey, eye; hy, hyoid crest segment; ma, mandibular crest segment; ot, otic vesicle. Unless otherwise noted, the mRNA doses of *HtrA1* and derived constructs per embryo were 100 pg.

*Figure 3 continued on next page*

Figure 3 continued

Indicated phenotypes were shown in B, 98/100; C, 74/83; D, 84/87; E, 31/31; F, 51/53; G, 59/60; H, 28/31; I, 23/26; J, 9/10; K, 21/21; L, 58/79 embryos; at least two independent experiments. Scale bars, 0.5 mm.

(*Figure 4C–D'*). The results suggest that HtrA1 overexpression and SerpinE2 knockdown impair with craniofacial cartilage formation.

Alcian Blue staining was used to visualize the cartilaginous skeleton in *Xenopus* tadpoles at stage 46 (*Figure 4E*). For quantification of the skeleton size, we measured the width at the level of the anterior ceratobranchial cartilages (*Figure 4—figure supplement 1A*). *HtrA1* mRNA injection reduced all cartilage structures and lowered the size of the head skeleton by 21% (*Figure 4F*, *Figure 4—figure supplement 1B and E*). Similarly, *SerpinE2*-MO decreased the skeleton size by 22% compared to control-morphant siblings (*Figure 4I and J*, *Figure 4—figure supplement 2A, B, and D*). The coiling pattern of the gut was not affected upon *HtrA1* mRNA injection (*Figure 4—figure supplement 1B*, arrowhead) and in *SerpinE2*-morphant tadpoles (*Figure 4—figure supplement 2B*, arrowhead), ruling out an overall delay of embryonic development. Co-injection of *SerpinE2*-MO together with non-targeted *Flag-SerpinE2* mRNA rescued craniofacial skeleton development (*Figure 4K*, *Figure 4—figure supplement 2C and D*). We conclude that HtrA1 and SerpinE2 regulate the formation of NC-derived cartilage structures.

## HtrA1 and SerpinE2 act in the NC to regulate cell migration and adherence to fibronectin

Since HtrA1 and SerpinE2 are expressed in both NC cells and surrounding tissue, we asked whether the proteins affect cell migration in an NC-autonomous or non-autonomous manner. The cranial NC can be dissected from *Xenopus* embryos at stage 17 and cultured on fibronectin in vitro to investigate collective cell migration in relative isolation, allowing for the identification of extrinsic versus intrinsic mechanisms (*Figure 5A*; *Alfandari et al., 2003*). After 4 hr, cells from uninjected explants spread as a coherent sheet toward one side and thereby doubled the surface area compared to the time of plating (*Figure 5B–B" and F*, filled arrowhead). In contrast, little cell migration was observed in explants upon injection with *HtrA1* mRNA, contributing to only a 20% increase in surface area within this time frame (*Figure 5C–C" and F*, open arrowhead). Similarly, *SerpinE2*-MO inhibited cell migration, leading to only one-tenth of the increase in total surface area that was observed in control-morphant explants (*Figure 5D–E" and F*). At 7 hr after plating, distinct segments were seen in uninjected and control-MO-injected explants (*Figure 5B‴ and D‴*), resembling the mandibular, hyoid, and branchial streams seen in sibling control embryos at the tailbud stage (*Figure 1D*). In contrast, explants injected with either *HtrA1* mRNA or *SerpinE2*-MO failed to display segmentation into distinct streams (*Figure 5C‴ and E‴*). In time-lapse video microscopy, control-MO-injected explants exhibited lamellipodia and filopodia at the leading front of the migratory NC cell clusters (*Figure 5—video 1*). *SerpinE2*-morphant NC cells lost polarity and took on a spherical appearance (*Figure 5—video 2*). These results show that both HtrA1 overexpression and SerpinE2 knockdown inhibit collective cell migration in vitro, providing evidence that HtrA1 and SerpinE2 regulate cell migration in the isolated NC.

To examine cell-matrix adhesion, we dissociated cranial NC explants in $Ca^{2+}/Mg^{2+}$-free medium and cultured them as single cells on fibronectin (*Figure 5G*). For better visibility of the NC cells, the donor embryos were injected with mRNA encoding enhanced green-fluorescent protein (eGFP). At 1 hr after plating, eGFP$^+$ cells adhered to fibronectin and extended cytoplasmic processes on this extracellular matrix protein (*Figure 5H*, filled arrowhead). Upon injection of *HtrA1* mRNA, nearly 80% of cells failed to attach and acquired a round morphology (*Figure 5I* and *Figure 5L*, open arrowhead). Similarly, *SerpinE2*-MO-injected cells lost adherence and cytoplasmic extensions, whereas the control-MO had no significant effect (*Figure 5J–L*). We conclude from these in vitro explant and single cell data that HtrA1 and SerpinE2 regulate in a NC-autonomous manner cell migration and adhesion on fibronectin.

## HtrA1 controls NC migration as a secreted protease

HtrA1 contains a cleavable N-terminal signal peptide (SP), a trypsin-like serine protease domain and a carboxyterminal PDZ (post-synaptic density of 95kD, discs large, zona occludens-1) protein-protein interaction domain (*Figure 6A*, top left; *Zurawa-Janicka et al., 2017*). The SP is a short hydrophobic

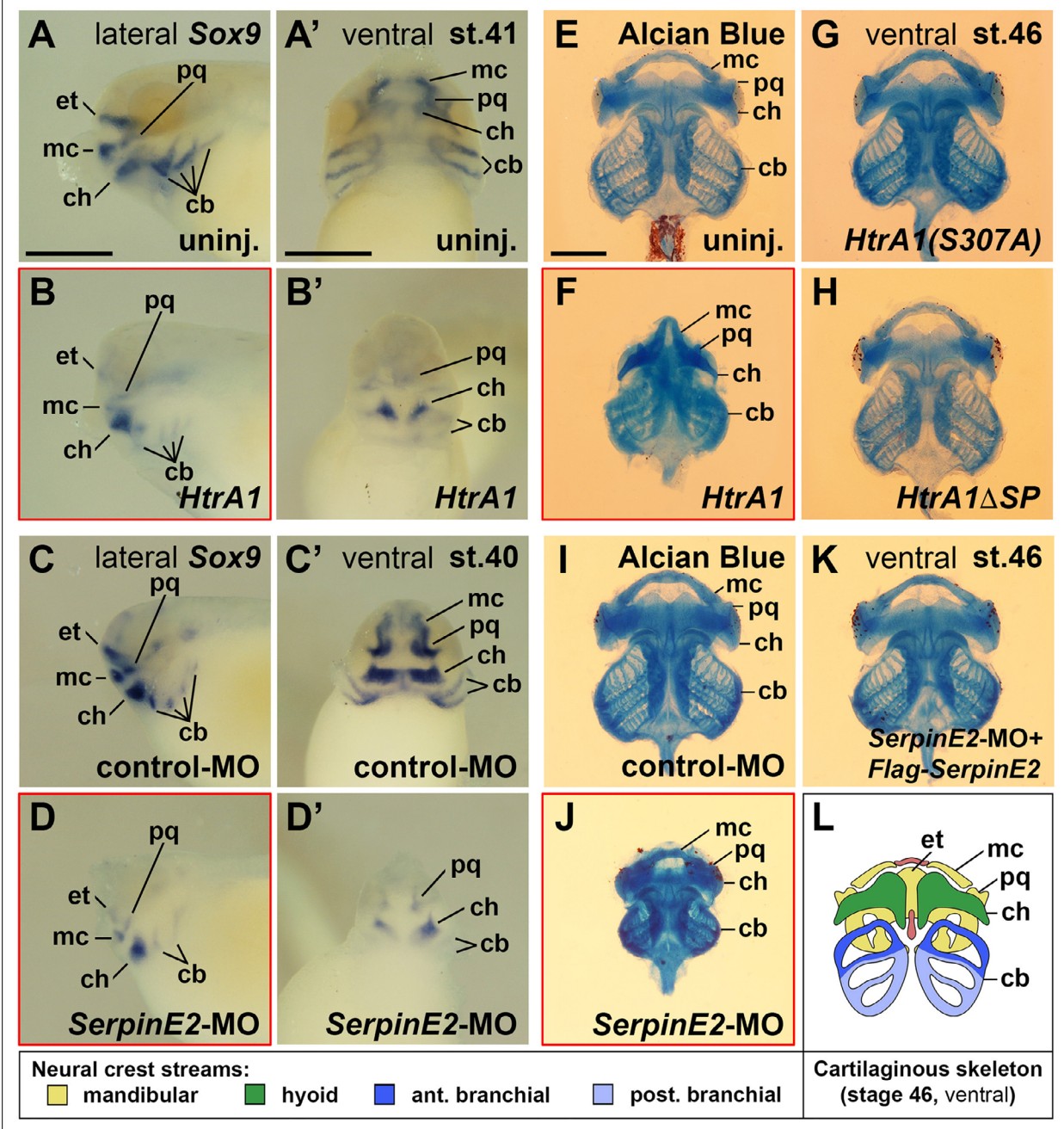

**Figure 4.** HtrA1 overexpression and SerpinE2 knockdown decrease cartilaginous elements and craniofacial structures. *Xenopus* embryos were injected into four animal blastomeres at the eight-cell stage with a total of 100 pg mRNA and 40 ng morpholino oligonucleotides (MOs). (**A–D'**) Tadpoles at stages 40/41 after whole-mount in situ hybridization in lateral (**A–D**) and ventral view (**A'–D'**). Note that *HtrA1* mRNA and *SerpinE2*-MO reduce *Sox9* expression, whereas control-MO has no effect on the labeled cartilaginous elements. (**E–K**) Ventral view of cartilaginous skeleton extracted from embryos at stage 46 after Alcian Blue staining. The dorsal ethmoid-trabecular cartilage was removed for better visibility. Note that *HtrA1* mRNA, but not *HtrA1(S307A)* and *Flag-HtrA1ΔSP* mRNAs, diminishes craniofacial structures (**E–H**). *SerpinE2*-MO, but not control-MO nor a combination of *SerpinE2*-MO and *Flag-SerpinE2* mRNA, reduce head skeleton structures (**I–K**). (**L**) Scheme of the cartilaginous skeleton at stage 46 in ventral view. Indicated is the contribution of neural crest streams to the craniofacial skeleton elements. cb, ceratobranchial; ch, ceratohyal; et, ethmoid-trabecular; mc, Meckel's cartilage; pq, palatoquadrate. Indicated phenotypes were shown in A, 11/11; B, 17/19; C, 12/13; D, 15/17; E, 57/58; F, 17/21; G, 72/77; H, 67/78; I, 83/88; J, 73/77; K, 77/87 embryos; at least two independent experiments. Scale bars, 0.5mm.

The online version of this article includes the following figure supplement(s) for figure 4:

**Figure supplement 1.** Overexpression of HtrA1 decreases cartilaginous elements and craniofacial structures.

**Figure supplement 2.** Knockdown of SerpinE2 reduces craniofacial skeleton formation.

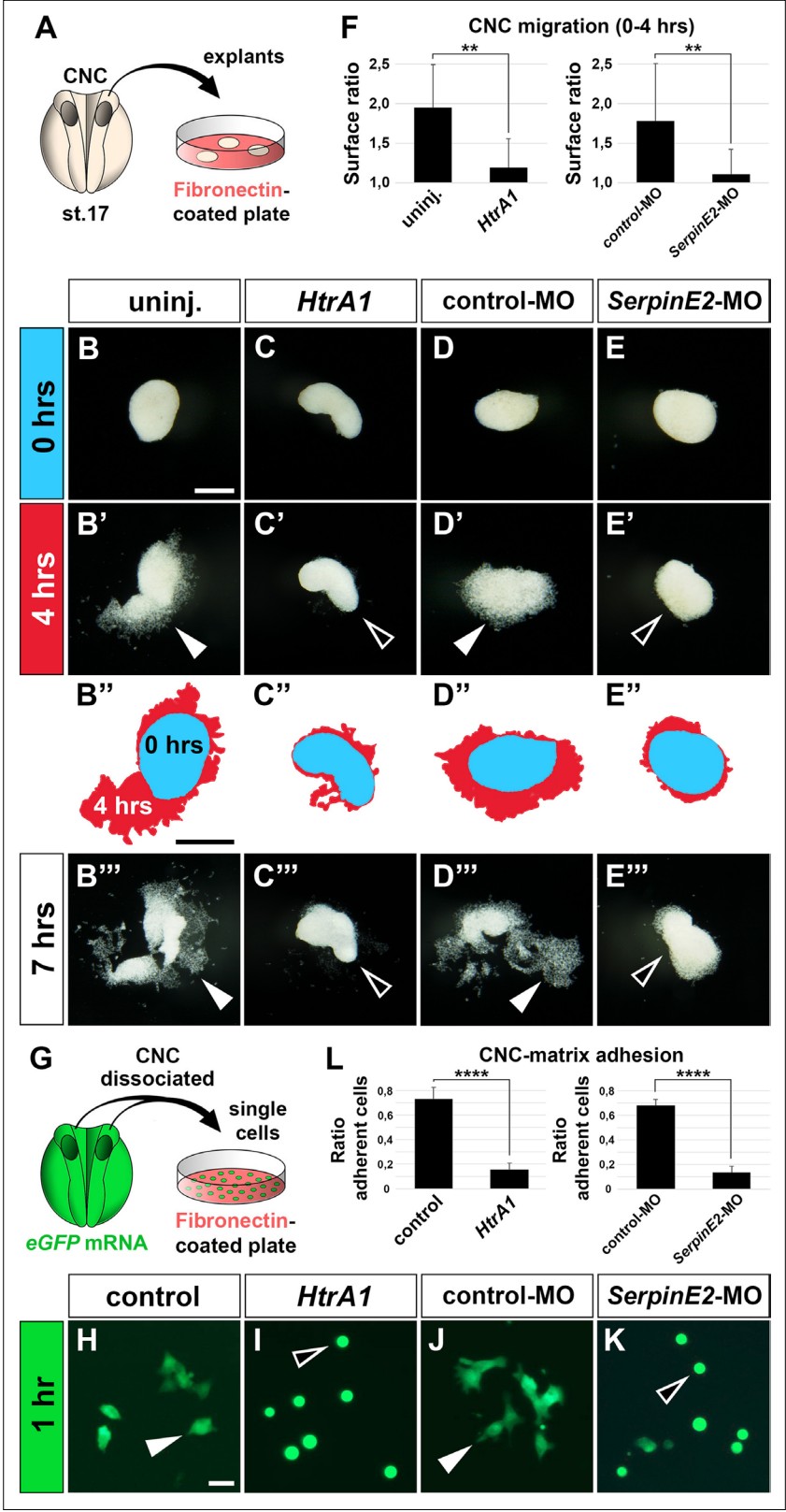

**Figure 5.** HtrA1 overexpression and SerpinE2 knockdown inhibit cranial neural crest cell migration and adhesion to fibronectin in vitro. (**A**) Scheme of migration experiment. The cranial neural crest was explanted from uninjected or injected embryos at stage 17 and cultured on a fibronectin-covered plastic plate. (**B–E'''**) Time-lapse of cell migration in CNC explants after culturing for 0, 4, or 7 hr. Note collective cell migration (filled arrowheads) in

*Figure 5 continued on next page*

*Figure 5 continued*

uninjected controls and explants injected with control-MO, whereas *HtrA1* mRNA and *SerpinE2*-MO block migration (open arrowheads). In B"–E", the surface areas of explants at 0 hr (blue) and 4 hr (red) were determined by ImageJ and superimposed. Scale bar, 0.2 mm. (**F**) Quantification of initial CNC migration. Indicated is the surface ratio of explants 4 hr versus 0 hr after plating. 12 explants were analyzed per sample. (**G**) Scheme of adhesion experiment. Upon injection of *eGFP* mRNA, CNC explants were dissociated in $Ca^{2+}$- and $Mg^{2+}$-free medium, and single cells were cultured on a fibronectin plate. (**H–K**) Single eGFP-labeled CNC cells after 1 hr culture. Note adhering cells with extended cytoplasmic processes (filled arrowheads) in control sample and after co-injection with control-MO, whereas *HtrA1* mRNA and *SerpinE2*-MO prevent adhesion causing injected cells to acquire a round phenotype (open arrowheads). Scale bar, 0.02 mm. (**L**) Quantification of CNC adhesion. Indicated is the ratio of adherent cells relative to the control. Analysis of n>1600 cells from at least six explants per sample. CNC, cranial neural crest; eGFP, enhanced green fluorescent protein. Embryos were injected with 100 pg mRNAs and 40 ng MOs. Data in all graphs are displayed as mean ± SD, n = 2; **p<0.01, ****p<0.0001, unpaired t-test.

The online version of this article includes the following video(s) for figure 5:

**Figure 5—video 1.** Collective migration of neural crest (NC) cells in vitro.

https://elifesciences.org/articles/91864/figures#fig5video1

**Figure 5—video 2.** Knockdown of SerpinE2 prevents adhesion and migration of neural crest (NC) cells.

https://elifesciences.org/articles/91864/figures#fig5video2

peptide sequence that destines proteins normally to the secretory pathway (*Owji et al., 2018*). We previously identified *Xenopus* HtrA1 as a protein in the supernatant of cDNA-transfected and metabolically labeled HEK 293T cells (*Hou et al., 2007*), suggesting that the protease might act in the extracellular space. On the other hand, human HtrA1 has been shown to associate with and stabilize microtubules in a PDZ domain-dependent manner (*Chien et al., 2009b*), raising the possibility for a non-proteolytic function of cytosolic HtrA1 in the regulation of cell motility. In a structure-function analysis, we investigated (1) whether HtrA1 acts as a protease in NC development, (2) whether HtrA1 operates in the extracellular space or inside the cell, and (3) whether its PDZ domain is involved in regulating NC migration. To this end, we compared with their cognate wild-type HtrA1 constructs three mutant derivatives, including HtrA1(S307A), in which the catalytic serine residue in amino acid position 307 is replaced by alanine (*Hou et al., 2007*), a newly generated Flag-tagged construct that lacks the secretory signal peptide (Flag-HtrA1ΔSP), and a myc-tagged construct that is devoid of the PDZ domain (HtrA1ΔPDZ-myc; *Acosta et al., 2015*; *Figure 6A*).

Using western blot analysis, we previously reported that HtrA1(S307A) (*Hou et al., 2007*) and HtrA1ΔPDZ-myc (*Acosta et al., 2015*) generate proteins of the expected sizes in mRNA-injected *Xenopus* embryos. These mutant HtrA1 constructs co-immunoprecipitate with Flag-SerpinE2 at protein levels similar to their corresponding wild-type HtrA1 and HtrA1-myc constructs (*Acosta et al., 2015*). Here, we used western blot analysis of lysates to show that the Flag-HtrA1 and Flag-HtrA1ΔSP protein constructs were expressed with the expected molecular weights in both cDNA-transfected HEK293T cells and mRNA-injected *Xenopus* embryos (*Figure 6—figure supplement 2A, B, and D*). However, while the Flag-HtrA1 protein accumulated at high levels in the supernatant of transfected cells, the Flag-HtrA1ΔSP protein failed to be efficiently secreted into the culture medium (*Figure 6—figure supplement 2C*). We also showed that wild-type Flag-HtrA1 and cytosolic Flag-HtrA1ΔSP degraded αTubulin, but not βActin, which validates the proteolytic activity and target specificity of these constructs (*Figure 6—figure supplement 2B and E–G*).

Neither *HtrA1(S307A)* nor *Flag-HtrA1ΔSP* affected NC migration in mRNA-injected embryos (*Figure 6D–E'*; see also *Figure 6—figure supplement 1A and B*). These two mutant constructs also failed to induce any defects in dorsal fin, melanocyte (*Figure 3C and D*) and craniofacial skeleton structures (*Figure 4G and H*; *Figure 4—figure supplement 1C–E*), suggesting that HtrA1 relies on an intact protease domain and transport to the secretory pathway to regulate the development of these NC derivatives. *HtrA1ΔPDZ-myc* efficiently inhibited EMT and NC migration to a degree comparable to that induced by *HtrA1-myc* control mRNA (*Figure 6F–G'*; *Figure 6—figure supplement 1A and B*). These results support the conclusions that HtrA1 acts as an extracellular protease and that an association of HtrA1 via its PDZ domain to microtubuli appears not to regulate the migratory behavior of NC cells in the *Xenopus* embryo.

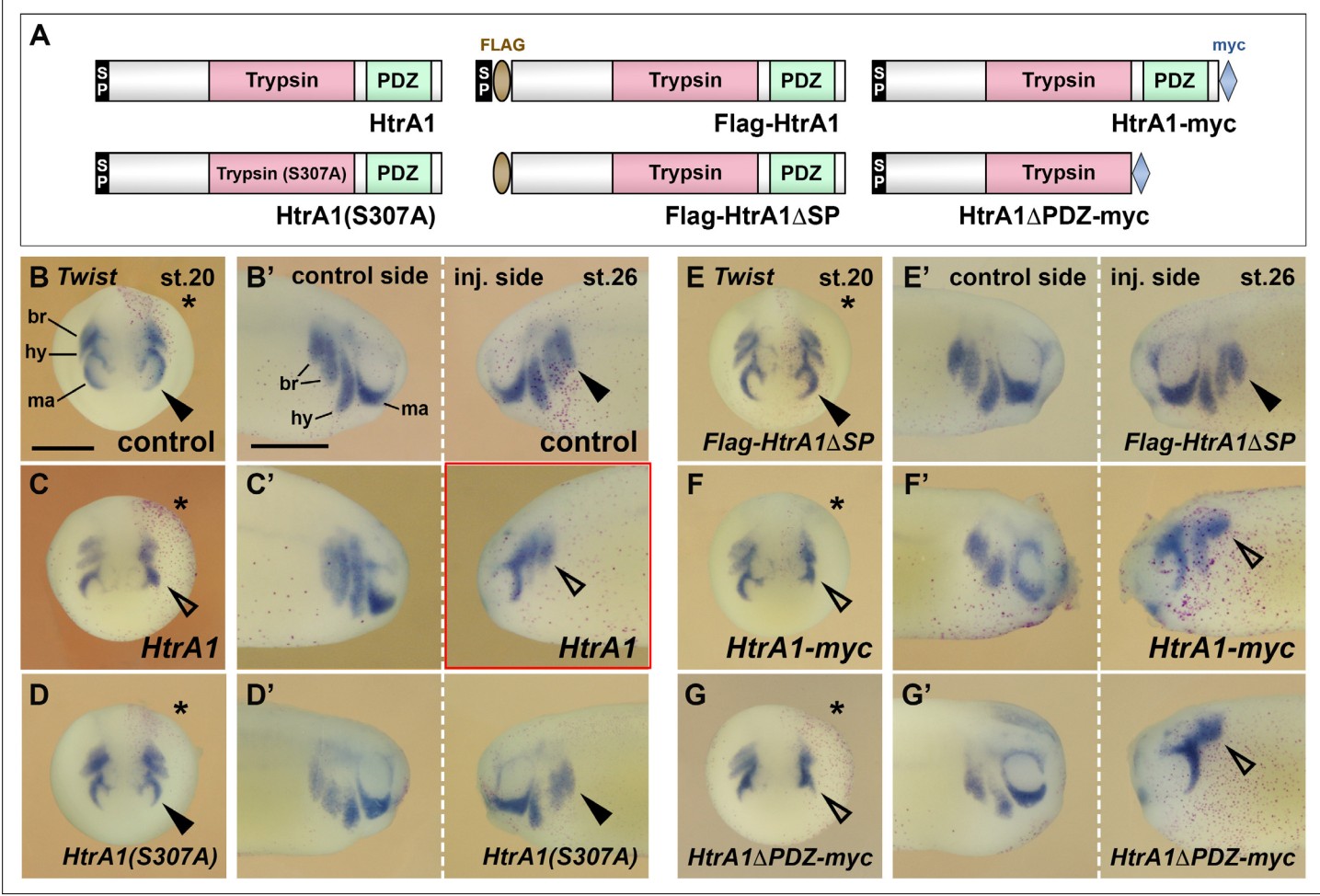

**Figure 6.** HtrA1 inhibits neural crest (NC) migration as an extracellular protease. Embryos were injected into a single dorsal animal blastomere at the eight-cell stage. A star labels the injected side. *Twist* expression demarcates the NC in embryos at stage 20 (B–G; anterior view) and stage 26 (B'–G'; lateral view). (**A**) Overview of wild-type (top) and mutant (bottom) HtrA1 protein constructs. (**B–E'**) *HtrA1* mRNA, but neither *HtrA1(S307A)* nor *Flag-HtrA1ΔSP* mRNAs, reduces epithelial-mesenchymal transition (EMT) and migration of NC cells on the injected side (arrowheads). Note that the diffusible HtrA1 protein reduces NC cell migration to a lower extent also on the non-injected side. (**F–G'**) Both *HtrA1-myc* and *HtrA1ΔPDC-myc* mRNAs reduce NC EMT and migration. br, branchial segments; hy, hyoid segment; ma, mandibular segment. Injected mRNA doses per embryos are 65 pg. Scale bars, 0.5mm. For quantification of NC migration defects, see *Figure 6—figure supplement 1A and B*.

The online version of this article includes the following source data and figure supplement(s) for figure 6:

**Figure supplement 1.** *HtrA1* mRNA inhibits neural crest (NC) migration in a concentration-dependent manner.

**Figure supplement 2.** Cytoplasmic HtrA1 causes reduction of αTubulin protein levels in mammalian cells and *Xenopus* embryos.

**Figure supplement 2—source data 1.** Uncropped western blots of HEK293T cell lysates after transfection of Flag-tagged HtrA1, HtrA1ΔSP, SerpinE2, and SerpinE2ΔSP cDNAs.

**Figure supplement 2—source data 2.** Uncropped western blots of HEK293T cell supernatants after transfection of Flag-tagged HtrA1, HtrA1ΔSP, SerpinE2, and SerpinE2ΔSP cDNAs.

**Figure supplement 2—source data 3.** Uncropped gels and western blots of *Xenopus embryo* lysates after injection of Flag-tagged HtrA1, HtrA1ΔSP, SerpinE2, and SerpinE2ΔSP mRNAs.

## SerpinE2 interacts with HtrA1 in NC cell migration

The serpin superfamily comprises extracellular and intracellular members with an exposed RCL that is cleaved by a target protease and irreversibly inhibits the attacking enzyme by forming a covalent serpin-protease complex (*Olson and Gettins, 2011*). SerpinE2 contains an N-terminal signal peptide and a C-terminal RCL (*Figure 7A*). We previously showed that SerpinE2 via its RCL binds to the trypsin domain of HtrA1 (*Acosta et al., 2015*). Do SerpinE2 and HtrA1 interact in NC cell migration? To this

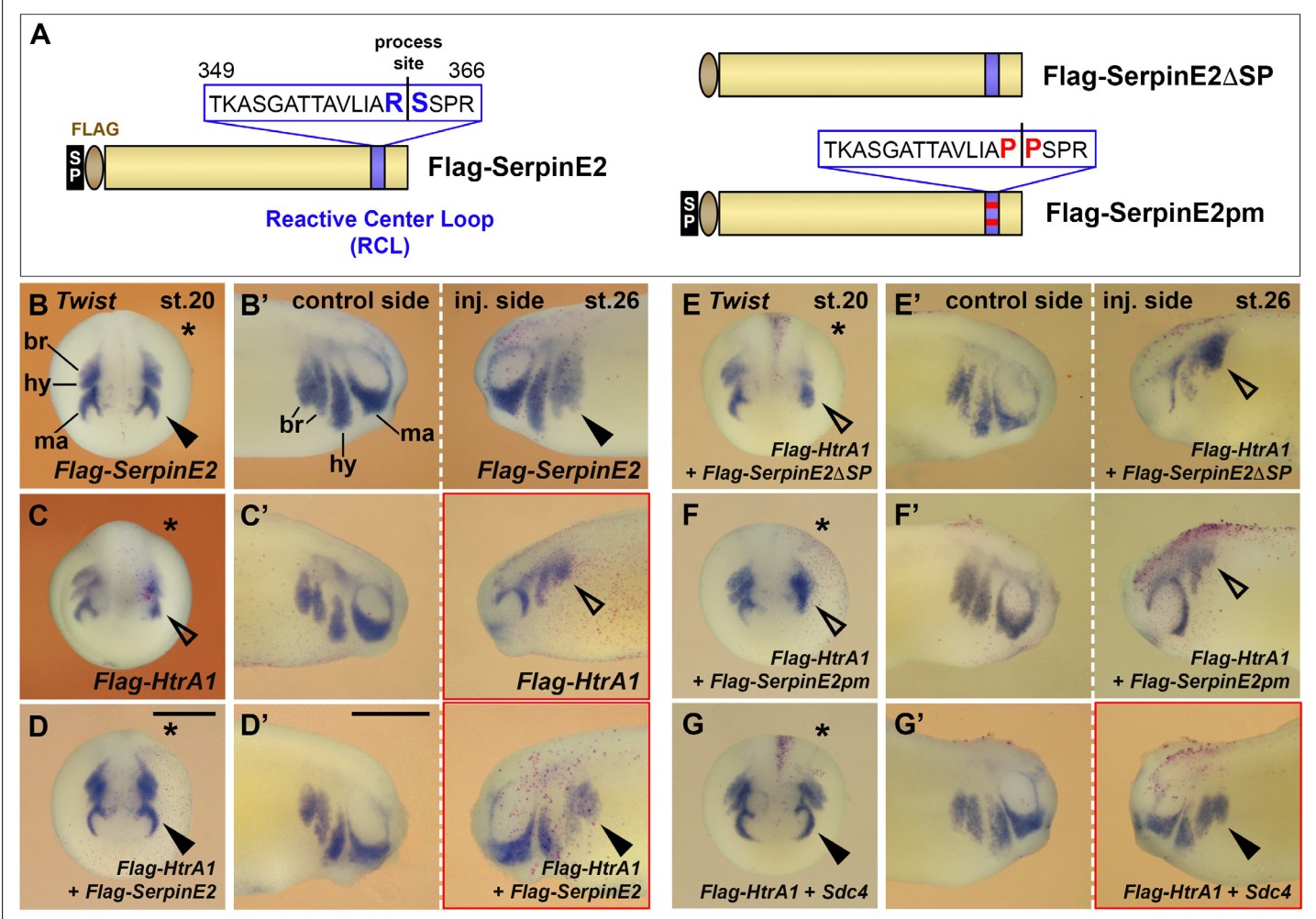

**Figure 7.** SerpinE2 and HtrA1 interact with Syndecan-4 (Sdc4) in neural crest (NC) cell migration. mRNAs were injected into one dorsal animal blastomere at the eight-cell stage. Embryos are shown in anterior view (stage 20, injected side labeled with a star, B–G) and lateral view (stage 26, B'–G'). (**A**) Overview of wild-type (left) and mutant (right) SerpinE2 protein constructs. (**B, B'**) 4 ng *Flag-SerpinE2* mRNA has no effect on the migration of *Twist*+ NC cells (filled arrowheads). (**C, C'**) Flag-HtrA1 inhibits NC cell migration robustly on the injected sides (open arrowheads). (**D–F'**) *SerpinE2* mRNA, but neither *Flag-SerpinE2ΔSP* nor *SerpinE2pm* mRNA, rescues normal epithelial-mesenchymal transition (EMT) and migration of NC cells upon co-injection with *Flag-HtrA1*. (**G, G'**) *Sdc4* mRNA restores normal NC migration in *Flag-HtrA1*-injected embryos. If not otherwise indicated, injected mRNA doses per embryos are 65 pg (*Flag-HtrA1*), 333 pg (*Flag-SerpinE2* derived constructs), and 450 pg (*Sdc4*). Scale bars, 0.5mm. For quantification of NC migration defects, see *Figure 7—figure supplement 1A and B*.

The online version of this article includes the following figure supplement(s) for figure 7:

**Figure supplement 1.** *SerpinE2* and *Sdc4* mRNAs partially rescue neural crest (NC) migration defects that are induced by HtrA1 overexpression.

end, we used Flag-tagged SerpinE2 (Flag-SerpinE2), a point mutant Flag-SerpinE2pm derivative, in which two proline residues replace the critical arginine and serine residues (R362P and S363P) at the process site of the RCL, and a newly generated truncated construct that lacks the N-terminal signal peptide (Flag-SerpinE2ΔSP) (*Figure 7A*). In previous western blot studies, we showed that Flag-SerpinE2 and Flag-SerpinE2pm are synthesized in similar protein amounts and at the expected sizes, but that overexpressed HtrA1 immunoprecipitates Flag-SerpinE2pm less efficiently than Flag-SerpinE2 (*Acosta et al., 2015*). Here, we use western blotting to validate that Flag-SerpinE2ΔSP protein is generated in expected amount and size upon mRNA injection in *Xenopus* embryos (*Figure 6—figure supplement 2A, D, and E*). However, unlike wild-type Flag-SerpinE2, the signal peptide-deficient Flag-SerpinE2ΔSP construct is not efficiently secreted in cDNA-transfected HEK293 cells (*Figure 6—figure supplement 2B and C*).

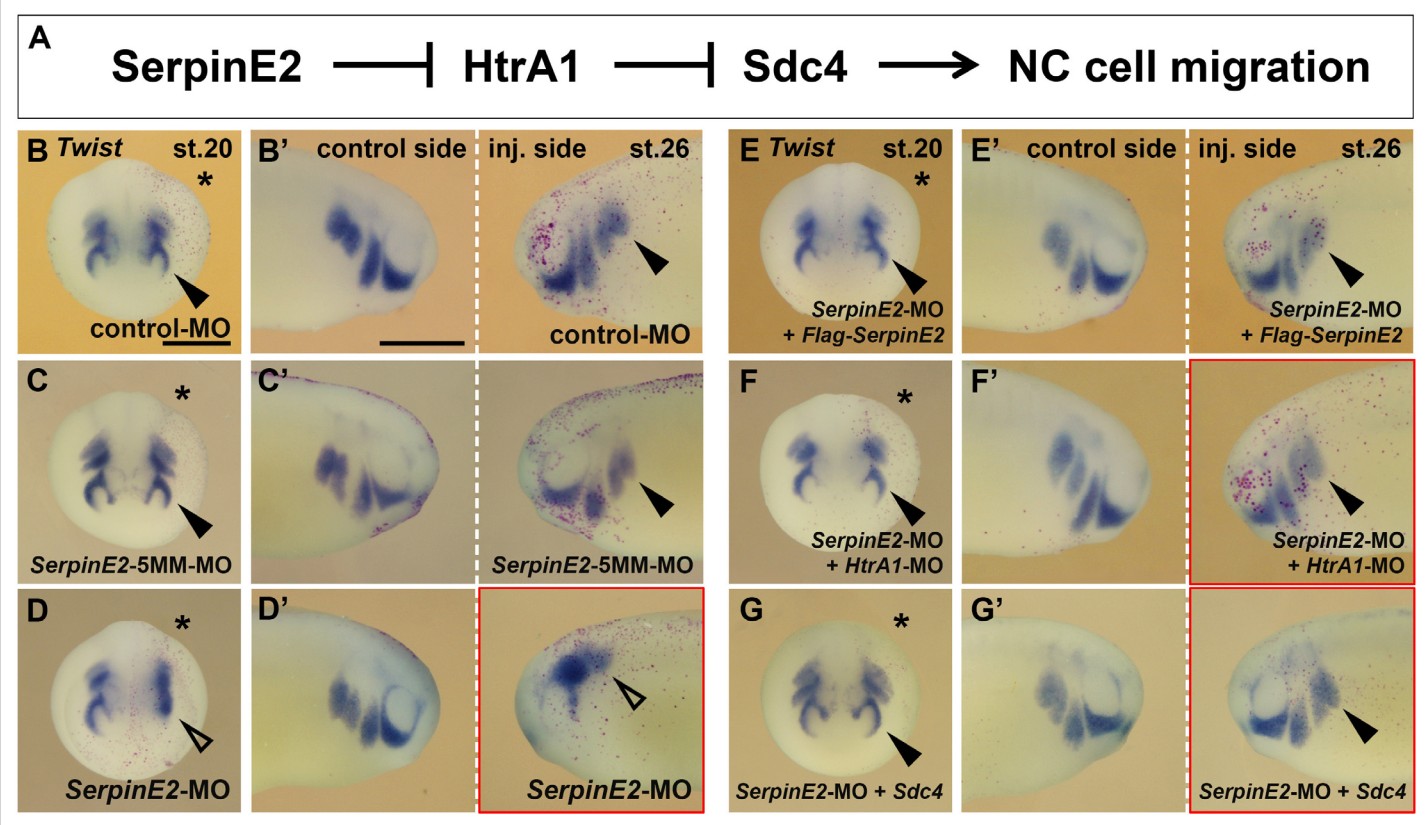

**Figure 8.** SerpinE2 functions in neural crest (NC) cell migration in an HtrA1- and Syndecan-4 (Sdc4)-dependent manner. mRNAs and morpholino oligonucleotides (MOs, 10 ng) were injected into one dorsal animal blastomere at the eight-cell stage. Embryos are shown in anterior view (stage 20, injected side labeled with a star, B–G) and lateral view (stage 26, B'–G'). (**A**) Proposed mechanism for the regulation of NC migration by SerpinE2, HtrA1, and Sdc4. (**B–D'**) *SerpinE2*-MO blocks epithelial-mesenchymal transition (EMT) and migration of *Twist*⁺ NC cells (arrow) on the injected side, while control-MO and *SerpinE2*-5MM-MO have no effect. (**E–G'**) *Flag-SerpinE2* mRNA, *HtrA1*-MO, and *Sdc4* mRNA restore normal NC migration in *SerpinE2*-morphant embryos. Injected mRNA doses per embryos are 333 pg (*Flag-SerpinE2*) and 450 pg (*Sdc4*). Scale bars, 0.5mm. For quantification of NC migration defects, see *Figure 8—figure supplement 1A and B*.

The online version of this article includes the following figure supplement(s) for figure 8:

**Figure supplement 1.** SerpinE2-MO causes neural crest (NC) migration defects which are rescued by *HtrA1*-MO and *Sdc4* mRNA.

Microinjection of *Flag-SerpinE2* mRNA at doses of up to 4 ng did not affect the migration of *Twist*⁺ NC cells (marked with nuclear lacZ lineage tracer) at stages 20 and 26 (*Figure 7B and B'*; see also *Figure 7—figure supplement 1*). However, co-injection of *Flag-HtrA1* and *Flag-SerpinE2* mRNAs reverted the EMT and NC migration defects that were induced by *Flag-HtrA1* mRNA alone (*Figure 7C–D'*; *Figure 7—figure supplement 1*), suggesting that SerpinE2 can relieve HtrA1-mediated suppression of NC migration. *Flag-SerpinE2ΔSP* failed to rescue the *Flag-HtrA1*-induced NC migration defects (*Figure 7E and E'*; *Figure 7—figure supplement 1*), underscoring that entry into the secretory pathway is important for the function of SerpinE2. Moreover, *Flag-SerpinE2pm* did not inhibit *Flag-HtrA1* mRNA from blocking NC migration (*Figure 7F and F'*; *Figure 7—figure supplement 1*). We conclude that SerpinE2 functions in the extracellular space and requires an intact RCL for efficient interaction with HtrA1 to modulate NC cell migration.

## Double knockdown of SerpinE2 and HtrA1 rescues NC migration

We next asked whether the NC migration defects induced by SerpinE2 depletion are dependent on endogenous HtrA1 protein. In loss-of-function experiments, we co-injected *SerpinE2*-MO and *HtrA1*-MO into a single dorsal blastomere and assessed the effects on NC cells. Importantly, knock-down of HtrA1 significantly reduced the EMT and migration defects in SerpinE2-depleted embryos (*Figure 8D, D', F, and F'*; see also *Figure 8—figure supplement 1A and B*). These epistatic

experiments support the existence of an extracellular proteolytic regulatory system, in which SerpinE2 stimulates NC cell migration through the inhibition of HtrA1.

## HtrA1 and SerpinE2 regulate cranial NC migration via Sdc4

Syndecans comprise a family of single-pass transmembrane proteoglycans with four members in vertebrates (*Keller-Pinter et al., 2021*). Sdc4 is a major component of focal adhesions and interacts with fibronectin during cell-matrix adhesion and cell movement (*Woods and Couchman, 2001*). We previously showed that HtrA1 triggers the proteolytic cleavage of Sdc4 and that SerpinE2 protects this proteoglycan from degradation by HtrA1 (*Hou et al., 2007*; *Acosta et al., 2015*). In *Xenopus* embryos, Sdc4 is abundant in the NC; knockdown of Sdc4 does not affect the induction of the NC but reduces its cell migration (*Matthews et al., 2008*). The effects of HtrA1 overexpression and SerpinE2 downregulation described in this study mimic the effects of Sdc4 depletion (*Matthews et al., 2008*). To investigate the relationship between the SerpinE2-HtrA1 axis and Sdc4 in the NC, we dorsally injected *Sdc4* mRNA and found it reduced in a concentration-dependent manner the EMT and NC migration defects induced by *Flag-HtrA1* mRNA (*Figure 7C, C', G, and G'*; see also *Figure 7—figure supplement 1A and B*). Ventral co-injection of *Sdc4* and *Flag-HtrA1* mRNAs was less efficient in rescuing NC cell migration (*Figure 7—figure supplement 1C and D*), because the transmembrane Sdc4 protein - unlike the secreted HtrA1 protease - remains anchored to ventral regions of the embryo and does not reach the dorsally located NC cells. Importantly, dorsally injected *Sdc4* mRNA partially restored normal EMT and cell migration in *SerpinE2*-morphant embryos (*Figure 8D, D', G and G'*; see also *Figure 8—figure supplement 1A and B*). Therefore, our epistatic studies in *Xenopus* embryos suggest that SerpinE2 promotes NC cell migration by inhibiting HtrA1-mediated degradation of Sdc4.

## Discussion

This study reveals, for the first time, a role for the SerpinE2 and HtrA1 proteolytic pathway in embryonic cell migration. Several lines of evidence support the conclusion that this inhibitor/protease pair functionally interact in the control of NC cell motility. (1) SerpinE2 and HtrA1 were co-expressed in pre-migratory and migrating NC cells in the *Xenopus* embryo. (2) In gain-of-function studies, wild-type HtrA1, but not a protease-defective construct, inhibited NC migration and development of NC-derived structures, such as branchial arch cartilage, dorsal fin tissue, and melanocytes. (3) SerpinE2 reverted the HtrA1-induced migration defects in mRNA-injected embryos. (4) In loss-of-function studies, SerpinE2 knockdown inhibited NC migration and development of NC-derived structures. (5) Concomitant knockdown of SerpinE2 and HtrA1 restored normal NC migration in MO-injected embryos. Additional epistatic experiments showed that *Sdc4* mRNA partially rescues the migration defects induced by HtrA1 overexpression or SerpinE2 knockdown. Thus, SerpinE2, HtrA1, and Sdc4 form an important regulatory axis to control NC development. SerpinE2 promotes NC cell migration by inhibiting endogenous HtrA1 and preventing this protease from degrading the transmembrane proteoglycan Sdc4 in the *Xenopus* embryo. Hence, our study reveals a critical role for the SerpinE2-HtrA1-Sdc4 axis of extracellular proteins to regulate collective cell migration in vivo (*Figure 8A*).

### Proteolytic control of morphogens

Secreted proteases, particularly of the astacin class of zinc metalloproteases, are important for the formation of morphogen gradients. In Hydra, the HAS-7 protease normally processes Wnt3 and restricts head organizer formation; its knockdown results in ectopic organizers (*Ziegler et al., 2021*; *Holstein, 2022*). In *Xenopus*, Tolloid regulates Spemann's organizer function by cleaving Chordin and de-repressing BMP signaling (*Piccolo et al., 1997*). The secreted Frizzled-related protein Sizzled binds to and inhibits Tolloid (*Lee et al., 2006*) and controls patterning of the dorsoventral axis through the following pathway:

Sizzled ⊣ Tolloid protease ⊣ Chordin ⊣ BMP

Our group previously showed that the serine protease HtrA1 induces mesoderm and ectopic tail formation by triggering the cleavage of Sdc4 and releasing active FGF messages (*Pera et al., 2005*). SerpinE2 (previously named PN1) binds to and inhibits HtrA1 (*Acosta et al., 2015*) and controls germ layer formation and patterning of the anteroposterior axis as follows:

## Extracellular proteases in cell migration

Migrating NC cells face major challenges to overcome physical barriers, such as the basal membrane or the extracellular matrix, suggesting proteolysis as an important mechanism for these cells to invade other tissues and reach their destined targets in the embryo. Matrix metalloproteases have been well studied in extracellular matrix remodeling during NC development (*Gouignard et al., 2018*). Less understood are other classes of proteases in the control of NC cell migration and invasion. The laboratory of Nicole Le Douarin was first to show that migrating NC cells produce serine proteases (*Valinsky and Le Douarin, 1985*). Using interspecific quail-chick grafting and enzyme-specific zymography, the group demonstrated high plasminogen activator activity in lysates of cranial NC compared to adjacent embryonic tissues.

Subsequent studies by other laboratories showed that mouse NC cells secrete urokinase and tissue plasminogen activators (uPA and tPA) into the culture medium (*Menoud et al., 1989*) and that uPA promotes chick NC migration in vitro via activation of plasmin and TGFβ signaling (*Brauer and Yee, 1993*; *Agrawal and Brauer, 1996*). Mutations in the lectin complement pathway gene MASP1/3, encoding for Mannose-associated serine protease-1 and -3, cause 3MC (Mingarelli, Malpuech, Michels and Carnevale) syndrome, a rare autosomal recessive disorder that is characterized by a spectrum of developmental features including craniofacial abnormalities (*Rooryck et al., 2011*). Zebrafish morphants exhibit craniofacial cartilage and pigment defects as well as abnormal NC migration, suggesting that MASP1 is a guidance cue that directs the migration of NC cells in the early embryo. Of note, all secreted serine proteases listed above have a positive role in NC migration. As we now demonstrate, HtrA1 is the only serine protease identified so far that acts as a negative regulator of NC migration.

SerpinE2 has a broad spectrum of target serine proteases that it binds to and inhibits, including uPA, tPA, and plasmin (*Arocas and Bouton, 2015*). Given the pro-migratory properties of these serine proteases, one should expect that SerpinE2 would inhibit NC migration by antagonizing their activity. However, overexpression of *SerpinE2* at mRNA doses of up to 4 ng did not affect the migration of NC cells in the *Xenopus* embryo. Instead, knockdown by MOs that block endogenous SerpinE2 protein biosynthesis (*Acosta et al., 2015*) efficiently inhibited NC migration and the development of NC-derived structures, as shown in this study. The finding that microinjection of *SerpinE2* mRNA reverted migration defects in these morphant embryos supports the view that SerpinE2 specifically promotes NC migration. It therefore appears that uPA, tPA, and plasmin are not target proteases of SerpinE2 in the modulation of NC migration in the *Xenopus* embryo.

## SerpinE2 and HtrA1 form a proteolytic pathway in NC migration

In *Xenopus* embryos at the neurula stage, *HtrA1* transcripts were most abundant in the superficial (ependymal) layer of the ectoderm containing non-motile epithelial cells, whereas *SerpinE2* expression was confined to the deep (sensorial) layer of the ectoderm that gives rise to motile mesenchymal NC cells (*Figure 1G*). In post-neurula embryos, transcripts of these genes appeared in the collective of migrating NC cells, with *SerpinE2* accumulating near the front of the cell streams and *HtrA1* being enriched at their rear ends (*Figure 1H*). We propose that only in regions where the SerpinE2 concentration is sufficiently high, HtrA1-mediated repression of cell motility is relieved so that NC cell migration can occur. In support of this conclusion, microinjection of *HtrA1* mRNA reduced in a concentration-dependent manner EMT and migration of NC cells, leading to defects in craniofacial skeleton structures. A key experiment was that EMT and migration of NC cells were restored by HtrA1 knockdown in SerpinE2-depleted embryos, suggesting that SerpinE2 promotes NC migration via inhibiting endogenous HtrA1 protease activity in the embryo. Our finding that *HtrA1* mRNA and *SerpinE2*-MO diminished cell migration in isolated NC explants provided evidence that the two proteins act in an NC-autonomous manner. HtrA1 overexpression and SerpinE2 knockdown also reduced *Sox9*-expressing chondrogenic precursors in the NC-derived head mesenchyme and caused defects in the cartilaginous skull and hyobranchial skeleton in *Xenopus* tadpoles. Interestingly, increased *HtrA1* expression has been detected in cranial sutures of mice with thyroid hormone-induced craniofacial disruptions (*Howie et al., 2016*). Since proper migration of NC cells is essential

for the formation of bones, cartilage, and soft tissue in the head (*Minoux and Rijli, 2010*), elevated HtrA1 levels might disturb NC migration not only in *Xenopus*, but also in mammalian embryos.

## SerpinE2 and HtrA1 regulate NC migration as extracellular proteins

The expression patterns were consistent with loss- and gain-of-function data that SerpinE2 promoted EMT and NC cell migration, whereas HtrA1 had the opposite effect. The actin cytoskeleton and microtubules play important roles in cell migration (*Seetharaman and Etienne-Manneville, 2020*). Two cytoskeletal proteins with functions in cell movement, i.e., fascin (actin bundling) and talin1 (regulation of actin assembly in focal adhesions), were previously identified as proteolytic substrates of HtrA1 (*An et al., 2010*). It has also been shown that the protease cleaves tubulins (*Chien et al., 2009a*) and that HtrA1 binds to and stabilizes microtubules via its PDZ domain (*Chien et al., 2009b*). However, the significance of the degradation of cytoskeletal proteins by HtrA1 remains unclear and it is unknown whether the association between HtrA1 and microtubules is important for cell motility. Here, we showed that wild-type HtrA1, but not a derived construct that was lacking a secretory signal peptide (HtrA1ΔSP), inhibited NC migration as well as craniofacial, dorsal fin, and melanocyte development. An HtrA1 construct with a deletion of the PDZ domain (HtrA1ΔPDZ) efficiently reduced EMT and migration of NC cells. The results strongly suggest that HtrA1 acts primarily as an extracellular protease during NC collective cell migration.

SerpinE2 shares with other members of the Serpin family an RCL that is cleaved by target proteases at the process site and forms a covalent acyl-enzyme complex leading to irreversible inhibition of the protease (*Olson and Gettins, 2011*; *Arocas and Bouton, 2015*). The introduction of prolines to residues P1 and P1' at the RCL cleavage site reduces the ability of SerpinE2 to physically interact with the catalytic trypsin domain of HtrA1 and inhibits its protease activity (*Acosta et al., 2015*). Here, we demonstrated that wild-type SerpinE2 restored normal EMT and migration of NC cells in *HtrA1* mRNA-injected embryos, whereas a point mutant SerpinE2 construct with the mutations Arg362Pro and Ser363Pro at P1 and P1' of the scissile site (SerpinE2pm) failed to show any effect. In addition, a SerpinE2 construct without a secretory signal peptide (SerpinE2ΔSP) did not revert the HtrA1-induced NC migration defects. The results underscore that SerpinE2 promotes NC migration as an extracellular protease inhibitor.

## The SerpinE2-HtrA1-Sdc4 axis functions in NC migration

Sdc4 is a central component of focal adhesion complexes that regulate cell-matrix adhesion and cell migration in cooperation with members of the integrin family of transmembrane proteins (*Keller-Pinter et al., 2021*). In zebrafish and *Xenopus* embryos, Sdc4 is expressed in migrating NC cells and promotes directional NC cell migration via regulation of the small GTPase Rac1 and activation of non-canonical Wnt signaling in the planar cell polarity pathway (*Matthews et al., 2008*). Similarly, human SDC4 favors cell migration and invasion through activation of Wnt5A signals in melanoma (*O'Connell et al., 2009*), indicating conserved signaling downstream of this proteoglycan in NC cell migration and malignant progression of an NC-derived cancer.

Several metalloproteases (MMP-3, -9, and -14) and serine proteases (plasmin, thrombin) cleave Sdc4 preferentially at the juxtamembrane site, so that its ectodomain can be released from the cell surface (*Manon-Jensen et al., 2013*). We previously showed that HtrA1 triggers the proteolytic cleavage of *Xenopus* Sdc4 (*Hou et al., 2007*). Whether HtrA1 directly cleaves this transmembrane protein or induces its cleavage through activation of other proteases remains to be shown. We further reported that SerpinE2 through physical interaction prevents HtrA1-mediated Sdc4 degradation, and that endogenous SerpinE2 is needed to protect the integrity of Sdc4 in embryos (*Acosta et al., 2015*). Here, we showed that *Sdc4* mRNA rescued NC migration defects that were induced by co-injection of *HtrA1* mRNA, indicating that Sdc4 is a relevant target of HtrA1 in vivo. The finding that *Sdc4* mRNA also rescued migration defects in *SerpinE2*-morphant embryos suggests a proteolytic pathway by the following double inhibition mechanism:

SerpinE2 ⊣ HtrA1 protease ⊣ Syndecan-4 → NC cell migration.

## SerpinE2 and HtrA1 might modulate FGF signals in directed NC migration

FGF signals including FGF8 stimulate NC migration by increasing cell motility and guiding cell movement (*Kubota and Ito, 2000*; *Sato et al., 2011*). In *Xenopus* embryos, the target tissues of NC streams secrete FGF8 and other FGF ligands, while the migrating NC cells express FGF receptors, including FGFR1 and FGFR4 (*Brändli and Kirschner, 1995*; *Lea et al., 2009*). Once released from the cells, FGFs are bound by heparan sulfate (HS) chains of cell surface proteoglycans such as the transmembrane Sdc4, which limit their diffusion (*Matsuo and Kimura-Yoshida, 2013*). We previously reported that the HtrA1-mediated proteolytic cleavage of Sdc4 mobilizes FGFs complexed to its soluble ectodomain, allowing these FGF-proteoglycan messages to activate FGFRs at distance (*Hou et al., 2007*). HS protects FGFs against cleavage by serine proteases (*Saksela et al., 1988*; *Sommer and Rifkin, 1989*), and HtrA1 can degrade free FGF8 protein (*Kim et al., 2012*). We showed that the HtrA1 inhibitor SerpinE2 stabilizes Sdc4 and regulates long-range FGF signaling in the *Xenopus* embryo (*Acosta et al., 2015*). The expression data presented here suggest that opposing gradients of SerpinE2 and HtrA1 form in the collective of migrating NC cells (*Figure 1H*). By controlling the mobilization of FGF signals, the SerpinE2/HtrA1 pair might help to establish a chemotactic FGF gradient and thereby facilitate collective migration of NC cells (*Figure 9B*).

## Fibronectin is a possible target of the SerpinE2-HtrA1 axis in NC cell adhesion and migration

Binding of fibronectin to the HS chains of Sdc4 and to the extracellular domain of the integrin adhesion receptors is critical for the activation of intracellular signaling that affects actin polymerization and contraction (*Keller-Pinter et al., 2021*). Fibronectin is ubiquitously expressed along NC migration pathways in the *Xenopus* embryo (*Davidson et al., 2004*). This glycoprotein is the only known extracellular matrix component that promotes *Xenopus* cranial NC cell migration as an adhesive substrate (*Alfandari et al., 2003*). Our study showed that HtrA1 overexpression and SerpinE2 knockdown reduced adhesion of cranial NC cells to fibronectin in vitro. Since fibronectin is a proteolytic substrate of HtrA1 (*Grau et al., 2006*; *Hadfield et al., 2008*), degradation of this matrix component might contribute to the HtrA1-mediated inhibition of NC migration.

## A gradient of serine protease activity may act in collective NC migration

We are proposing a mechanism, in which SerpinE2 and HtrA1 constitute a proteolytic pathway that regulates collective NC migration by targeting the focal adhesion protein Sdc4 and the matrix protein fibronectin (*Figure 9A*). Our structure-function analyses and epistatic experiments led us suggest a model in which SerpinE2 and HtrA1 regulate directed migration of the NC collective via remodeling of cell-matrix adhesions (*Figure 9C*). Since SerpinE2 has a high affinity for HS through its heparin binding domain (*Li and Huntington, 2012*), the transmembrane Sdc4 proteoglycan likely recruits this protease inhibitor to the pericellular space. Elevated SerpinE2 concentration keeps HtrA1 activity low near the leading edge of the NC collective; this protects the integrity of both Sdc4 and fibronectin in focal adhesions, which is critical for the leader cells to attach to the matrix and drive cell migration (left side in *Figure 9C*). Gradually decreasing SerpinE2 protein levels and a concomitant increase in HtrA1 protease activity behind the leader cells triggers degradation of Sdc4 and cleavage of fibronectin, causing disruption of focal adhesion complexes in follower cells and loss of cell-matrix binding at the rear end of the NC cluster (right side in *Figure 9C*).

## SERPINE2, HTRA1, and SDC4 might interact in human placenta development and pre-eclampsia

An interaction of SerpinE2, HtrA1, and Sdc4 in cell migration might not be confined to the NC as shown in this study but could also occur in other aspects of development and disease. In the developing placenta, extravillous trophoblast (EVT) cells from the human embryo migrate into the inner uterus wall (endometrium) and invade the uterine spiral arteries, converting them into large blood vessels so that the blood flow to the embryo is enhanced. Inadequate EVT migration results in insufficient maternal artery remodeling, causing hypoxia of the placenta and hypertension in a pathological

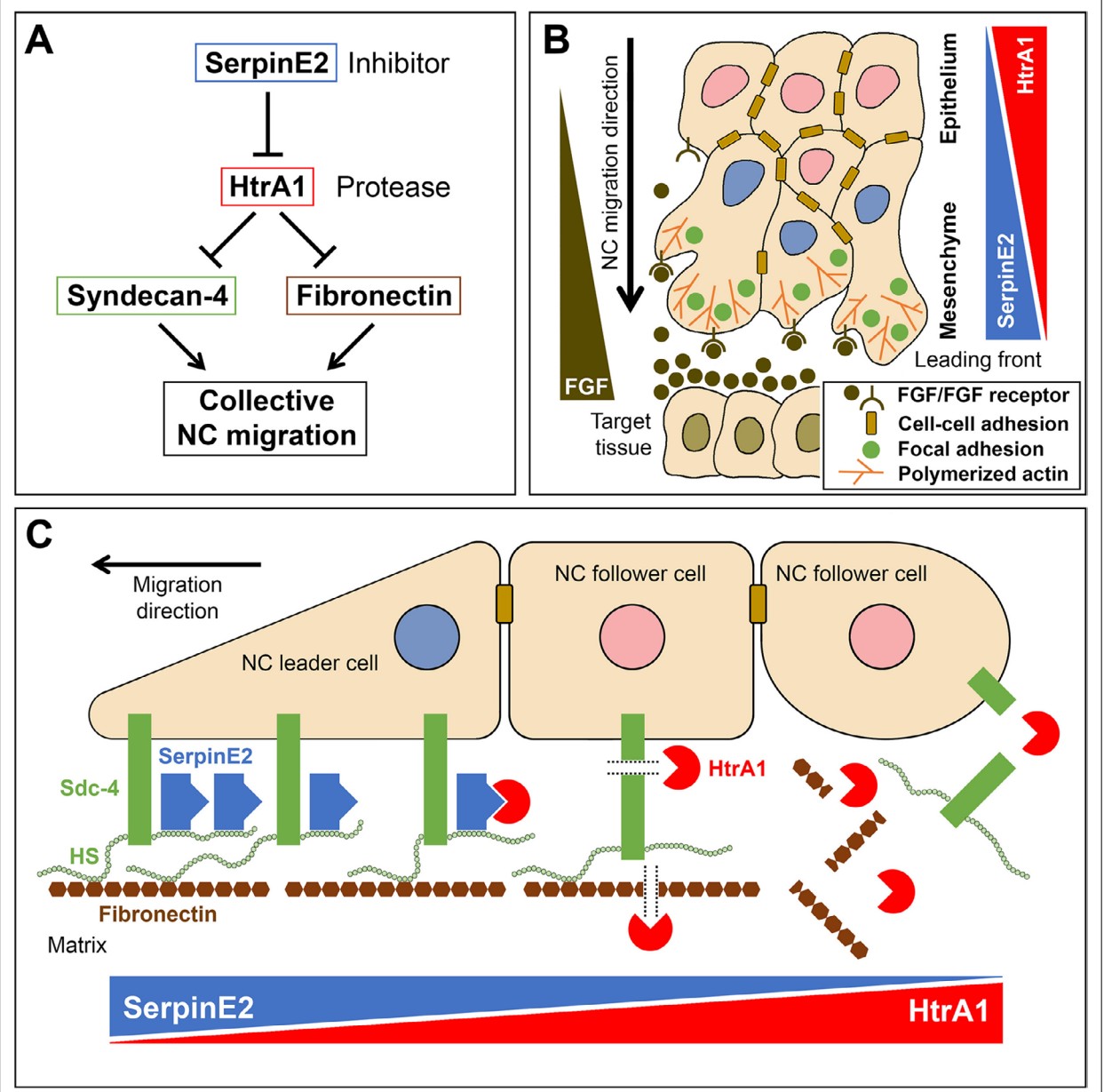

**Figure 9.** Model for a proteolytic pathway of SerpinE2 and HtrA1 that regulates collective neural crest migration. (**A**) SerpinE2 stimulates collective NC migration by a double-inhibitory mechanism involving the secreted serine protease HtrA1 and its proteolytic substrates Syndecan-4 and fibronectin. (**B**) Opposing gradients of SerpinE2 and HtrA1 activities regulate the directed migration in a collective of NC cells. The SerpinE2/HtrA1 pair contributes to the formation of a chemoattractant gradient that guides the NC stream toward a source of FGF signals in the target tissue. High SerpinE2 and low HtrA1 levels coincide with abundant focal adhesion sites and polymerized actin that drive mesenchymal migration at the leading edge. (**C**) SerpinE2 anchored to the heparan sulfate chains of the transmembrane proteoglycan Syndecan-4 protects the integrity of focal adhesions at the leading front and allows collective cell migration to occur (left side). Unbound HtrA1 triggers the proteolytic cleavage of Syndecan-4 and degrades the matrix protein fibronectin (middle), causing loss of cell-matrix adhesion at the rear end of the NC cell collective (right side). FGF, fibroblast growth factor; HS, heparan sulfate; NC, neural crest; Sdc4, Syndecan-4.

condition called pre-eclampsia that is potentially life-threatening for both fetus and mother. The placenta expresses the highest levels of SERPINE2 among all probed human tissues (The Human Protein Atlas; https://www.proteinatlas.org/), with abundant protein expression in migratory EVTs and spiral arteries (*Chern et al., 2011*). HTRA1 expression is low in EVTs, high in less motile villous trophoblasts and upregulated in the placenta of patients with pre-eclampsia (*Ajayi et al., 2008*). Interestingly, the migration and invasion of cultured EVTs is inhibited by silencing of SERPINE2 (*Chern et al.,*

2011), overexpression of HTRA1 (*Ajayi et al., 2008*), and knockdown of SDC4 (*Jeyarajah et al., 2019*). The results suggest that the SERPINE2-HTRA1-SDC4 axis might regulate trophoblast motility during placental development and a serious pregnancy condition in humans.

## The SERPINE2-HTRA1-SDC4 regulatory axis might act in cancer

Cell migration is a hallmark of tumor malignancy and essential for metastasis formation (*Hanahan and Weinberg, 2011*). SERPINE2 is upregulated in aggressive variants of several cancer types, and high expression levels correlate with poor prognosis in patients (*Arocas and Bouton, 2015*). Downregulation of SERPINE2 impairs the metastatic spread of melanoma and breast cancer cells in xenografted mice (*Wagenblast et al., 2015*; *Wu, 2016*; *Tang et al., 2019*), suggesting a positive role of SERPINE2 in cancer cell migration and metastasis. On the other hand, HTRA1 is downregulated in many primary tumors and metastatic foci, while low levels correlate with poor clinical outcome (*Zurawa-Janicka et al., 2017*). Overexpression of HTRA1 inhibits the invasion and migration of melanoma and breast cancer cells, respectively (*Baldi et al., 2002*; *Wang et al., 2012*). SDC4 is overexpressed in several tumor types such as melanoma, and high expression levels are associated with poor prognosis in breast cancer (*Keller-Pinter et al., 2021*). Gene silencing confirms the requirement of SDC4 in the metastasis of human breast carcinoma cells in mice (*Leblanc et al., 2018*). Together, the double-inhibitory mechanism involving the molecular interaction of SERPINE2, HTRA1, and SDC4 as suggested here for NC cell migration in the *Xenopus* embryo might also regulate cancer cell migration in human metastasis.

## Conclusions

This is the first study to demonstrate an interaction of SerpinE2 and HtrA1 in cell migration in vivo. We showed that SerpinE2 and HtrA1 acted in the extracellular space at least partly through the cell surface proteoglycan Sdc4 to govern NC migration in *Xenopus* embryos. The secreted SerpinE2 and HtrA1 proteins, as well as the ectodomain of Sdc4, are amenable to neutralizing antibodies. Our structure-function analysis unraveled important roles for the RCL of SerpinE2 and the catalytic serine residue of its target protease, whereas the PDZ domain of HtrA1 was not involved in the control of NC cell migration. The identification of critical domains in SerpinE2 and HtrA1 provides an important basis for the development of effective therapeutics in craniofacial anomalies and other neurocristopathies. Given the roles of these proteins in trophoblast migration in the developing placenta, together with their implication in metastasis of numerous tumor types, the SERPINE2-HTRA1-SDC4 axis constitutes a potential therapeutic target for the treatment of pre-eclampsia and various aggressive cancers.

# Materials and methods

## Key resources table

| Reagent type (species) or resource | Designation | Source or reference | Identifiers | Additional information |
|---|---|---|---|---|
| Cell line (*Homo sapiens*) | HEK293T | TakaraBio | Cat# 632180 | Used for transient transfection and protein production |
| Biological sample (*X. laevis*, male) | Sperm | Nasco | Cat# LM00715M | Used for in vitro fertilization |
| Biological sample (*X. laevis*, male) | Sperm | *Xenopus* 1 | Cat# 5215 | Albino; used for in vitro fertilization |
| Biological sample (*X. laevis*, female) | Egg | Nasco | Cat# LM00535M | Used for in vitro fertilization |
| Biological sample (*X. laevis*, female) | Egg | Nasco | Cat# LM00510M | Albino; used for in vitro fertilization |
| Biological sample (*X. laevis*, female) | Egg | *Xenopus* 1 | Cat# 4280 | Used for in vitro fertilization |
| Antibody | Anti-flag M2-Peroxidase (HRP) (Mouse monoclonal) | Sigma-Aldrich | Cat# A8592 | WB (1:1000) |
| Antibody | Anti-αTubulin (Mouse monoclonal) | Sigma-Aldrich | Cat# T5168 clone B-5-1-2 | WB (1:1000) |
| Antibody | Anti-βActin (Mouse monoclonal) | Sigma-Aldrich | Cat# A5441 clone AC-15 | WB (1:10,000) |

*Continued on next page*

*Continued*

| Reagent type (species) or resource | Designation | Source or reference | Identifiers | Additional information |
|---|---|---|---|---|
| Antibody | Anti-HtrA1 (Rabbit polyclonal) | *Hou et al., 2007* PMID:17681134 | Immuno-purified GST5057, G22-7 | WB (1:2500) |
| Antibody | Mouse IgG HRP (Goat polyclonal) | R&D Systems | Cat# AF007 | WB (1:2000) |
| Antibody | Rabbit IgG HRP (Goat polyclonal) | R&D Systems | Cat# HAF008 | WB (1:2000) |
| Commercial assay or kit | NucleoBond PC 100, Midi Kit | Macherey-Nagel | Cat# 740573.100 | DNA purification |
| Commercial assay or kit | NucleoSpin Gel and PCR Clean-up, Mini kit | Macherey-Nagel | Cat# 740609.50 | DNA purification |
| Commercial assay or kit | mMessage mMachine SP6 Transcription kit | Invitrogen | Cat# AM1340 | mRNA synthesis |
| Commercial assay or kit | RNeasy Mini Kit | QIAGEN | Cat# 74104 | RNA purification |
| Commercial assay or kit | Pierce BCA Protein Assay Kit | Thermo Fisher Scientific | Cat# 23225 | Protein measurement |
| Chemical compound, drug | Pfu DNA Polymerase | Thermo Fisher Scientific | Cat# EP0572 | DNA synthesis |
| Chemical compound, drug | Lipofectamin 3000 Transfection Reagent | Thermo Fisher Scientific | Cat# L3000001 | Transfection |
| Chemical compound, drug | RIPA Lysis and Extraction Buffer | Thermo Fisher Scientific | Cat# 89901 | Protein purification |
| Chemical compound, drug | Halt Protease and Phosphatase Inhibitor Cocktail | Thermo Fisher Scientific | Cat# 78442 | Protein purification |
| Chemical compound, drug | cOmplete, EDTA-free Protease Inhibitor Cocktail | Roche | Cat# 11873580001 | Protein purification |
| Chemical compound, drug | Bolt Bis-Tris Plus Mini Protein Gels, 4–12% | Thermo Fisher Scientific | Cat# NW04125BOX | Protein electrophoresis |
| Chemical compound, drug | Bolt Sample Reducing Agent (10×) | Thermo Fisher Scientific | Cat# B0009 | Protein electrophoresis |
| Chemical compound, drug | Bolt LDS Sample Buffer (4×) | Thermo Fisher Scientific | Cat# B0007 | Protein electrophoresis |
| Chemical compound, drug | Bolt MES SDS Running Buffer (20×) | Thermo Fisher Scientific | Cat# B0002 | Protein electrophoresis |
| Chemical compound, drug | PageRuler Prestained Protein Ladder, 10–180 kDa | Thermo Fisher Scientific | Cat# 26616 | Protein electrophoresis |
| Chemical compound, drug | Ponceau S Solution | Sigma-Aldrich | Cat# P7170 | Western blotting |
| Chemical compound, drug | Pierce ECL Western Blotting Substrate | Thermo Fisher Scientific | Cat# 32106 | Western blotting |
| Chemical compound, drug | Restore PLUS Western Blot Stripping Buffer | Thermo Fisher Scientific | Cat# 46430 | Western blotting |
| Chemical compound, drug | Human Plasma Fibronectin Purified Protein | Sigma-Aldrich | Cat# FC010 | Neural crest explant culture |
| Chemical compound, drug | Gentamycin Solution | Sigma-Aldrich | Cat# G1272 | Embryo culture |
| Software, algorithm | ImageJ | NIH https://imagej.net/ij/ | RRID:SCR_003070 | Neural crest explant measurement |
| Software, algorithm | AxioVision 4.8 | Zeiss https://www.zeiss.com/ | RRID:SCR_002677 | Neural crest explant imaging |
| Other | 35 mm Dish, No. 0 Coverslip, 10 mm Glass Diameter, uncoated | MatTek | Cat# P35G-0-10C | Neural crest explant culture, time-lapse imaging |
| Other | Vivaspin 2 MWCO 10,000 | Cytiva | Cat# 28932247 | Protein concentration |
| Other | iBlot Transfer Stack, PVDF, mini | Thermo Fisher Scientific | Cat# IB401002 | Western blotting |

## Materials availability statement

Requests for resources and further information should be directed and will be fulfilled by the corresponding author, Edgar M Pera.

## Constructs

pCS2-*Flag-HtrA1* (identical with pCS2-*xHtrA1\**; *Hou et al., 2007*) and pCS2-*Flag-SerpinE2* (identical with pCS2-*Flag-PN1*; *Acosta et al., 2015*) were previously described. Briefly, N-terminally truncated

open reading frames of *X. laevis* HtrA1.S (amino acids 17–459) and SerpinE2.L (amino acids 20–395) were each introduced into a modified pCS2 expression vector in frame and downstream of a secretion cassette with the sequence <u>MQCPPILLVWTLWIMAVDC</u>SRPKVFLPIQPEQEPLQSKT(DYKDDDDK) LE that contains a cleavable signal peptide (underlined) and N-terminus until Thr39 of *X. laevis* Chordin, followed by a Flag-tag (in brackets) and two amino acids (LE) representing an *Xho*I cloning site (pCS2-Chd*SP-Flag*, constructed by Stefano Piccolo in the laboratory of Eddy De Robertis, UCLA). Using pCS2-*Flag-HtrA1* and pCS2-*Flag-SerpinE2* as templates, the N-terminally truncated open reading frames of HtrA1 and SerpinE2 were PCR-amplified, using a forward primer that inserts a start codon before the N-terminal Flag-tag, and subcloned into the expression vector pCS2 to generate pCS2-*Flag-HtrA1ΔSP* and pCS2-*Flag-SerpinE2ΔSP*. The PCRs were performed with high fidelity *Pfu* DNA polymerase (Thermo Fisher, EP0572), and correct sequences were validated by sequencing in sense and antisense directions (Eurofins, Germany). Protein bands were quantified using Image Lab (Bio-Rad).

## Antisense morpholino oligonucleotides (purchased from Gene Tools LLC)

| Name | Forward | Reference |
|------|---------|-----------|
| Standard control-MO | 5'- CCT CTT ACC TCA GTT ACA ATT TAT A | Gene Tools LLC |
| *SerpinE2.L* (*PN1.a*)-MO1 | 5'- GAA GTC AAG TAA GAA TAC TCC CGG C | *Acosta et al., 2015* |
| *SerpinE2.L* (*PN1.a*)-MO2 | 5'- ACT AGT CGC CTC ATG ATC GTA CAA C | *Acosta et al., 2015* |
| *SerpinE2.S* (*PN1.b*)-MO | 5'- CAT GAT CGT AGA ACT GGA TAG AAG T | *Acosta et al., 2015* |
| *SerpinE2-5MM*-MO | 5'-ACT ACT CAC CTA ATG ATA GTA AAA C | *Acosta et al., 2015* |
| *HtrA1*-MO | 5'- ACA CCG CCA GCC ACA ACA TGG TCA T | *Hou et al., 2007* |

## *Xenopus* embryo microinjection

To prepare mRNA, pCS2 constructs containing *HtrA1*, *HtrA1-S307A*, and *Flag-HtrA1* (*Hou et al., 2007*), *HtrA1-myc*, *HtrA1ΔPDZ-myc*, *Flag-SerpinE2* and *Flag-SerpinE2pm* (*Acosta et al., 2015*), *Flag-HtrA1ΔSP* and *Flag-SerpinE2ΔSP* (this study), *nlacZ* (a kind gift from Dr. Tomas Pieler, University Göttingen, Germany), *eGFP* (a kind gift from Dr. Eric Bellefroid, Université Libre de Bruxelles, Belgium) and *Flag-Sdc4* (*Muñoz et al., 2006*) were linearized with NotI and transcribed with Sp6 RNA polymerase (mMessage mMachine, Invitrogen). Embryos were injected in 1× MBS (Modified Barth's saline) and cultured in 0.1× MBS. Pigmented embryos were injected into all four animal blastomeres at the eight-cell *Muñoz et al., 2006* stage. Albino embryos were injected into a single animal blastomere with *nlacZ* mRNA as a lineage tracer and stained with Magenta-Red (Sigma B8931, red nuclei) after fixation. Dorsally and ventrally injected embryos were identified based on the lineage tracer pattern. Embryos were prepared, fixed, and processed by whole-mount in situ hybridization as described (*Pera et al., 2015*).

## Neural crest explantation, in vitro culture, and time-lapse imaging

For NC extirpation, the vitelline membrane was removed with fine forceps from pigmented embryos at stage 17 and the neural folds were extracted in 1× MMR (Marc's Minimal Ringers) buffer, using eye lashes mounted with nail polisher to pipette tips. Operated embryos were cultured in 0.3× MMR and 50 µg/ml Gentamycin (Sigma G1272) until stage 41.

CNC explants and single fluorescently labeled cells from *eGFP* mRNA-injected embryos were cultured in Danilchik's for Amy media (53 mM NaCl, 5 mM Na$_2$CO$_3$, 4.5 K gluconate, 32 mM Na gluconate, 1 mM MgSO$_4$, 1 mM CaCl$_2$, 0.1% bovine serum albumin, adjusted with 1 M bicine to pH 8.3) in fibronectin-coated plastic dishes as described (*Gouignard et al., 2016*).

For video time-lapse imaging, CNC explants were cultured in 23% Leibovitz's L15 medium in fibronectin-coated glass-bottom culture dishes (MatTek Corp. P35G-0-10C). The video was taken after 7 hr in culture. Filming was done with a Zeiss inverted microscope, and AxioVision 4.8.2 SP3 software was used for imaging.

## Cell culture and transfection

HEK293T cells were cultured in DMEM (Dulbecco's modified Eagle medium) supplemented with 10% heat-inactivated fetal bovine serum and penicillin/streptavidin. The cells were seeded at a density of 20–30% in Opti-MEM (Thermo Fisher Scientific) with penicillin/streptavidin in six-well plates, and transient transfection with plasmid DNAs was performed using Lipofectamin 3000 (Thermo Fisher Scientific), when cells reached between 60% and 70% confluency. Following a medium exchange 6 hr after transfection, the supernatant was harvested, and the cells were lysed 70 hr after transfection.

## Sample preparations and western blotting

Microinjected embryos and transfected HEK293T cells were lysed using RIPA-buffer (Thermo Fisher Scientific, Cat# 89901) supplemented with protease/phosphatase inhibitor cocktail (Thermo Fisher Scientific, Cat# 78442). The cell lysate was sonicated, using the Sonicator Bioruptor Plus (Diagenode) with 40 cycles for 15 s on and 15 s off. The supernatant was cleared from cells by centrifugations first at 2500 rpm and then at 8000 rpm for each 5 min at 4°C. After addition of a protease inhibitor cocktail (Roche, Cat# 11873580001), proteins in the supernatant were concentrated, using Vivaspin2 10 kDa molecular weight cut-off columns (Cytiva, Cat# 28932247). Proteins were measured in the lysates and supernatants, using the Pierce BCA Protein Assay Kit (Thermo Fisher Scientific, Cat# 23225). Gel electrophoresis was performed with 20 µg protein per lane, using Bolt Bis-Tris Plus Mini Protein gels (4–12%, Thermo Fisher Scientific, Cat# NW04125BOX). Western blots were performed, using the following primary antibodies: anti-flag HRP-conjugated (1:1000; Sigma-Aldrich, A8592), anti-αTubulin (1:1000; Sigma-Aldrich, T5168), anti-βActin (1:10,000; Sigma-Aldrich, A5441), immunopurified anti-HtrA1 (5 µg/ml; *Hou et al., 2007*).

## Statistical analysis

Results are shown as mean ± standard deviation (SD). Unpaired t-test was employed for comparing means of two treatment groups. For multiple comparisons, one-way ANOVA followed by Tukey's post hoc test was used. Statistical significance was defined as **$p<0.01$, ***$p<0.001$, and ****$p<0.0001$.

## Acknowledgements

We are indebted to Drs. E Bellefroid, EM De Robertis, J Larraine, J-P Saint Jeannet, and T Pieler for plasmids, to Drs. Erika Velásquez, Isak Martinsson, Oxana Klementieva, Gunnar Gouras for sharing advice and equipment, and to J Monka for language improvement of the manuscript. EMP wishes to thank Dr. Eddy De Robertis for valuable comments on the manuscript, suggestion of the neural crest extirpation experiment, generous support, and being a superb host during a memorable 3-month sabbatical stay (May to July 2023) in his laboratory at UCLA in Los Angeles (USA). Funding: This paper was supported by the following grants: Swedish Research Council 2009-4951 to Edgar M Pera. Swedish Childhood Cancer Fund. PROJ11/101 to Edgar M Pera. OE and Edla Johansson foundation to Edgar M Pera. Albert Påhlsson foundation to Edgar M Pera. Pia Ståhl foundation to Edgar M Pera. MultiPark to Laurent Roybon. Swedish Research Council 2021-02284 to Laurent Roybon.

## Additional information

### Funding

| Funder | Grant reference number | Author |
| --- | --- | --- |
| Swedish Research Council | 2009-4951 | Edgar M Pera |
| Swedish Childhood Cancer Fund | ProJ11/101 | Edgar M Pera |
| O. E. och Edla Johanssons Vetenskapliga Stiftelse | | Edgar M Pera |
| Albert Påhlsson Foundation | | Edgar M Pera |

| Funder | Grant reference number | Author |
|---|---|---|
| Pia Stahl Foundation | | Edgar M Pera |
| Swedish Research Council | 2021-02284 | Laurent Roybon |
| Lund University | MultiPark | Laurent Roybon |

The funders had no role in study design, data collection and interpretation, or the decision to submit the work for publication.

## Author contributions

Edgar M Pera, Conceptualization, Resources, Formal analysis, Supervision, Funding acquisition, Validation, Investigation, Visualization, Methodology, Writing - original draft, Project administration, Writing - review and editing; Josefine Nilsson-De Moura, Conceptualization, Formal analysis, Validation, Investigation, Visualization; Yuriy Pomeshchik, Resources, Formal analysis, Validation, Investigation; Laurent Roybon, Resources, Funding acquisition; Ivana Milas, Conceptualization, Formal analysis, Validation, Investigation, Methodology

## Author ORCIDs

Edgar M Pera (iD) http://orcid.org/0000-0002-0625-0006
Josefine Nilsson-De Moura (iD) http://orcid.org/0009-0008-9408-6189
Yuriy Pomeshchik (iD) http://orcid.org/0000-0002-1412-7403
Laurent Roybon (iD) http://orcid.org/0000-0002-5532-4964

## Ethics

This study was conducted in strict accordance with all local, national, and European regulations and ARRIVE guidelines. All Xenopus laevis experiments reported in this study were approved by the Lund/Malmö; regional ethical committee (Dnr. 5.8.18-15884/2019).

Reviewer #1 (Public Review): https://doi.org/10.7554/eLife.91864.3.sa1
Reviewer #2 (Public Review): https://doi.org/10.7554/eLife.91864.3.sa2
Author response https://doi.org/10.7554/eLife.91864.3.sa3

# Additional files

## Supplementary files

• MDAR checklist

## Data availability

All data generated or analysed during this study are included in the manuscript and supporting files. Three source data files have been provided for *Figure 6—figure supplement 2*.

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
