## [Editor Report · eLife assessment]

This **fundamental** work substantially advances our understanding of cell migration, especially in that of cranial neural crest. The additional evidence provided to support the conclusion is **exceptional**, with rigorous biochemical assays for materials used and with intensive genetic interventions. The work will be of broad interest to developmental biologists and cell biologists.

---

## [Referee Report · Reviewer #1 (Public Review)]

Summary:

A novel serine protease and a inhibitor pair regulate cell migration in the neural crest.

Strengths:

The reproduction of classical cranial neural crest extirpations and their phenocopy by SerpinE2 morpholino are remarkable. Very scholarly written and data of the highest quality.

Weaknesses:

All were improved upon revision.

---

## [Referee Report · Reviewer #2 (Public Review)]

Summary:

The authors conducted research on the role of SerpinE2 and HtrA1 in neural crest migration using *Xenopus* embryos. The data presented in this study was of high quality and supported the authors' conclusions. The discovery of the potential molecular connection between SerpinE2 and HtrA1 in neural crest cell migration in vivo is significant, as understanding this pathway could potentially lead to treatments for aggressive cancers and pregnancy-related disorders.

Strengths:

Previous research has shown that SerpinE2 and HtrA1 can have both positive and negative effects on cell migration, but their molecular interplay and role in neural crest migration are not well-established. This study is the first to reveal a potential connection between these two proteins in neural crest cell migration in vivo. The authors found that SerpinE2 promotes neural crest migration by inhibiting HtrA1. Additionally, overexpression of Sdc4 partly alleviates neural crest migration issues caused by SerpinE2 knockdown or HtrA1 overexpression. These findings suggest that the SeprinE2-HtrA1-Sdc4 pathway is crucial for neural crest migration.

Weaknesses:

To further increase the study's credibility, it may be helpful to use techniques like western blotting, qRT-PCR, or in situ hybridization to verify the efficiency of SerpinE2 and HtrA1 knockdown and/or overexpression.

---

## [Author Response]

The following is the authors’ response to the original reviews.

We thank the two reviewers for their very thoughtful suggestions and the editors for writing the eLife assessment. We will submit a revised manuscript that addresses most comments and include a point-by-point response to the reviewers. We will provide evidence that overexpression of the HtrA1 protease and knockdown of its inhibitor SerpinE2 reduce the development of neural crest-derived cartilage elements in the head of *Xenopus* embryos. This will be done by whole mount in situ hybridization, using a probe for the chondrogenic marker Sox9. We will also provide two time-lapse movies showing (1) collective migration of cranial neural crest cells in culture and (2) failure of these cells to adhere to fibronectin upon SerpinE2 depletion. We will discuss in more depth how the SerpinE2-HtrA1 proteolytic pathway and its target, the heparan sulfate proteoglycan Syndecan-4, might regulate FGF signaling and suggest a model, in which serpin secreted by the leader cells and the protease released by the follower cells might establish a chemotactic FGF gradient for the directed migration of the neural crest cohort. The criticism that other factors such as proliferation and cell survival might contribute to the observed craniofacial phenotypes upon misexpression of SerpinE2 and HtrA1, and that it remains unclear to what extent the mechanism reported here is conserved in the trunk neural crest is valid. The reason we focused on the more amenable cranial neural crest in the *Xenopus* embryo and used a multitude of approaches – structure-function studies, biochemical analyses, in vitro explant assays and epistatic experiments in vivo – was to validate a central finding: that an extracellular proteolytic pathway involving a serpin, a protease and a proteoglycan regulates by a double inhibition mechanism collective cell migration.